# PhiNets: Brain-inspired Non-contrastive Learning Based on Temporal Prediction Hypothesis

**Satoki Ishikawa**[1,2,*]**, Makoto Yamada**[2,*,†]**, Han Bao**[3,2]**, Yuki Takezawa**[2,3]
[1]Science Tokyo, [2]Okinawa Institute of Science and Technology, [3]Kyoto University

## Abstract

Predictive coding has been established as a promising neuroscientific theory to describe the mechanism of information processing in the retina or cortex. This theory hypothesises that cortex predicts sensory inputs at various levels of abstraction to minimise prediction errors. Inspired by predictive coding, Chen et al. (2024) proposed another theory, *temporal prediction hypothesis*, to claim that sequence memory residing in hippocampus has emerged through predicting input signals from the past sensory inputs. Specifically, they supposed that the CA3 predictor in hippocampus creates synaptic delay between input signals, which is compensated by the following CA1 predictor. Though recorded neural activities were replicated based on the temporal prediction hypothesis, its validity has not been fully explored. In this work, we aim to explore the temporal prediction hypothesis from the perspective of self-supervised learning (SSL). Specifically, we focus on non-contrastive learning, which generates two augmented views of an input image and predicts one from another. Non-contrastive learning is intimately related to the temporal prediction hypothesis because the synaptic delay is implicitly created by StopGradient. Building upon a popular non-contrastive learner, SimSiam, we propose *PhiNet*, an extension of SimSiam with two predictors explicitly corresponding to the CA3 and CA1, respectively. Through studying the PhiNet model, we discover two findings. First, meaningful data representations emerge in PhiNet more stably than in SimSiam. This is initially supported by our learning dynamics analysis: PhiNet is more robust to the representational collapse. Second, PhiNet adapts more quickly to newly incoming patterns in online and continual learning scenarios. For practitioners, we additionally propose an extension called X-PhiNet integrated with a momentum encoder, excelling in continual learning. All in all, our work reveals that the temporal prediction hypothesis is a reasonable model in terms of the robustness and adaptivity.

 https://github.com/riverstone496/PhiNets

## 1 Introduction

How does learning and adaptivity emerge in a biological system? It has been a long-standing question in both neuroscience and machine learning. In the neuroscience community, predictive coding has been a promising hypothesis to support the flexibility of biological brains. While predictive coding was initially proposed to explain cortical functions, Chen et al. (2024) recently extended predictive coding to propose the *temporal prediction hypothesis*, which claims that the hippocampus predicts future sensory inputs based on past experiences (Mumford, 1992; Rao and Ballard, 1999; Friston, 2005). Specifically, Chen et al. (2024, Figure 1) modelled the hippocampus with the CA3 predictor followed by the CA1 predictor—while the former yields synaptic delay to input signals, the latter compensates the time difference between the past and incoming signals by temporal prediction. This is the first attempt to explain the mechanism of sequence (short-term) memory from the viewpoint of temporal prediction. While they tested the temporal prediction hypothesis by using recorded neural activities, the validity of the hypothesis has not been explored sufficiently.

---

[*]Equal contribution
[†]Corresponding author, e-mail:makoto.yamada@oist.jp

This work is aimed at exploring a learning model built upon the temporal prediction hypothesis to see when the hypothesis is reasonable. To this end, we shed light on self-supervised learning (SSL). SSL is a paradigm to train a learner from input sensory patterns without supervised signals, which aligns to biological learning more closely. Over the past decade, machine learning researchers have developed a number of SSL models. Popular SSL models are SimCLR (Chen et al., 2020a) and MoCo (Chen et al., 2020b), which are contrastive learning methods that learn data representations with two augmented views generated from an input image by minimizing the InfoNCE loss (van den Oord et al., 2018), requiring a tremendous number of negative samples to stably obtain representations. Thus, we focus on another SSL model, non-contrastive learning, which learns data representations from only the two augmented views without requiring negative samples. Specifically, SimSiam (Chen and He, 2021) is a natural model to study the temporal prediction hypothesis—SimSiam predicts one augmented view of an input image from another view, introducing an implicit time difference through the StopGradient operation. For this reason, we choose SimSiam, unlike the other non-contrastive models such as Barlow Twins (Zbontar et al., 2021). This implicit connection between SimSiam and the temporal prediction hypothesis is an appealing test bed to computationally verify how memory-based prediction processes in the brain behave in different scenarios.

Building upon SimSiam, we propose the brain-inspired SSL model *PhiNet* for investigating the effectivity of the temporal prediction hypothesis in the context of machine learning. PhiNet extends SimSiam by incorporating an additional predictor after the original predictor. We associate the original and additional predictors with the CA3 and CA1 regions in the hippocampal model (see Figure 1 in Chen et al. (2024) and Figure 1b). We leverage PhiNet as a computational model to implement the temporal prediction hypothesis and study when it effectively learns sensory inputs.

Our first discovery is that PhiNet is less prone to the representational collapse, which leads to stable learning. Non-contrastive learning intrinsically faces the challenge of collapsing into a trivial representation because it eliminates explicit negative signals. In Section 4, we theoretically analyse the learning dynamics of PhiNet to reveal that PhiNet is less sensitive to initialization and the weight decay hyperparameter, and has a wider retraction basin to a non-trivial representation (in (C1)), relative to SimSiam. This supports the empirically better linear probing performance and hyperparameter robustness of PhiNet across different image datasets. Our second discovery is that PhiNet empirically performs better in online and continual learning, in particular. We tested PhiNet and baseline non-contrastive learners by using the CIFAR-5m dataset (Nakkiran et al., 2021), exposing learners to a gigantic amount of input images but with only significantly fewer epochs. In this scenario, effective memory functions are necessary to lead the learning to success. As a result, PhiNet exhibits better accuracy with less forgetting than SimSiam. Therefore, the effectiveness of the temporal prediction hypothesis is witnessed from the perspective of the robustness and adaptivity.

For practically better performance, we extend PhiNet to additionally propose *X-PhiNet*, which incorporates a momentum encoder, inspired by the Complementary Learning Systems (CLS) theory (McClelland et al., 1995). This extra momentum encoder represent long-term memory in the neocortex, storing information derived from the hippocampal model. X-PhiNet maintains good performances especially in online and continual learning scenarios.

**Contributions.**

- Section 3: we propose a new non-contrastive learning model called PhiNet, which is inspired by a hippocampal model (Chen et al., 2024).
- Section 4: we compare the learning dynamics (Tian et al., 2021) of PhiNet and SimSiam. Consequently, it elucidates that PhiNet can avoid the complete collapse of representations (Liu et al., 2023; Bao, 2023) more easily than SimSiam with the aid of the additional predictor.
- Section 5.1: we investigate the image classification performance of PhiNet using CIFAR and ImageNet datasets. We show that PhiNet performs comparably to SimSiam but is more robust against weight decay.
- Section 5.2: we further extend PhiNet by proposing X-PhiNet to integrate the neocortex model based on the Complementary Learning Systems (CLS) theory (McClelland et al., 1995). Experimentally, X-PhiNet works effectively in online and continual learning.

**Limitations**    One major limitation of our approach is the use of backpropagation, which differs from the mechanisms in biological neural networks. Our long-term goal is to eliminate backpropagation

to better imitate brain function, but this work focuses on structural aspects of network architectures. Currently, backpropagation-free predictive coding mechanisms for complex architectures like ResNet are in the early stages of development, with most research being limited to simple CNNs. Future research should explore if the proposed structure can enable effective learning with backpropagation-free predictive coding. Another key difference between PhiNet and brains is the presence of recurrent structures. Making the data into time series data and adding a recurrent structure to the model remains as future work.

## 2 RELATED WORK

**Brain-inspired methods.** Predictive coding, initially introduced as a theory of the retina (Srinivasan et al., 1982), has gained attention as a unifying theory of cortical functions (Mumford, 1992; Rao and Ballard, 1999; Friston, 2005). They suggest that brains operate by predicting sensory inputs at various levels of abstraction to minimise prediction errors. Recent studies have leveraged these ideas for contrastive learning (van den Oord et al., 2018; Henaff, 2020). Chen et al. (2024) extended the predictive coding theory to the hippocampus with the temporal prediction hypothesis. Specifically, the temporal prediction hypothesis supposes that prediction errors are calculated with the CA1 model and used to update the CA3 model. Some studies have attempted to apply the hippocampal model to representation learning (Pham et al., 2021; 2023). Among them, DualNet refines representation learning based on CLS theory (McClelland et al., 1995; Kumaran et al., 2016), which supposes that the interplay between slow (self-supervised) and fast (supervised) architectures is the basis of brain learning. Pham et al. (2021) examined supervised learning tasks alongside self-supervised training.

**Self-supervised learning.** Current mainstream approaches to self-supervised learning (SSL) often rely on cross-view prediction frameworks (Becker and Hinton, 1992), with contrastive learning emerging as a prominent SSL paradigm. In contrastive learning like SimCLR (Chen et al., 2020a), a network contrasts positive (similar) and negative (dissimilar) samples to learn data representations. One limitation of SimCLR is its empirical reliance on gigantic negative samples. Theoretically, contrastive learning essentially requires huge number of negative samples (Bao et al., 2022; Awasthi et al., 2022). To address this issue, recent research has focused on approaches free from negative sampling (Grill et al., 2020; Caron et al., 2020; 2021). For instance, BYOL (Grill et al., 2020) trains representations by aligning online and target networks, where the target network is created by maintaining a moving average of the online network parameters. SimSiam (Chen and He, 2021) utilises a Siamese network to align two augmented views of an input by fixing one of the networks with StopGradient. While the lack of negative samples may easily yield collapsed representations, namely, constant representations, Tian et al. (2021) analysed the BYOL/SimSiam dynamics with a two-layer network and found that complete collapse is prevented unless weight decay is excessively strong. We partially leverage their analysis framework to explain the mechanism of our PhiNet. In recent years, many studies have leveraged SimSiam for continual learning (Smith et al., 2021; Madaan et al., 2022) and reinforcement learning (Tang et al., 2023). RM-SimSiam (Fu et al., 2024) and CaSSLe (Fini et al., 2022) enhance the performance of continual learning by incorporating a memory block into SimSiam, while its architecture has not been neuroscientifically grounded. In video self-supervised learning, built on predictive coding principles, a common strategy is to train a predictor that takes a frame or clip from one time-step to generate a distinct representation at a different time-step (Han et al., 2019; 2020; Tan et al., 2023; Bardes et al., 2024).

Note that our aim is to bridge the temporal prediction hypothesis and self-supervised learning. To this end, *non-contrastive* learning provides a better model because both hippocampus and neocortex do not have any mechanism corresponding to negative sample generation. Specifically, SimSiam is a simple yet powerful learning model, and we can benefit from its StopGradient to effectively draw a connection to predictive coding. Thus, we focus on SimSiam as a backbone model in this work.

## 3 PHINETS (Φ-NETS)

In this paper, we propose PhiNets, which are non-contrastive methods based on CLS theory (McClelland et al., 1995) and the temporal prediction hypothesis (Chen et al., 2024). Chen et al. (2024, Figure 1) provides a hippocampal model, where the entorhinal cortex (EC) serves as an input signal layer, the CA3 region serves as the predictor, and the CA1 region measures the prediction error.

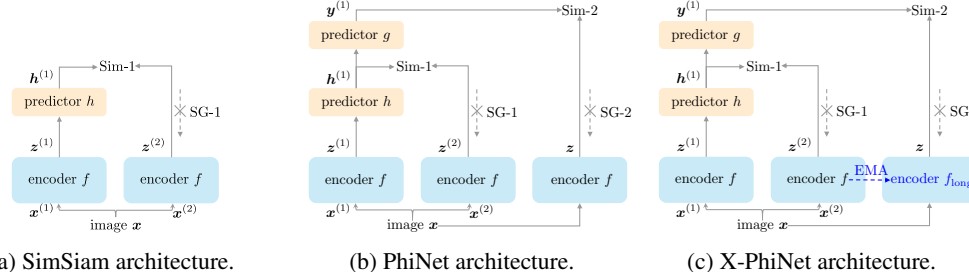

(a) SimSiam architecture.     (b) PhiNet architecture.     (c) X-PhiNet architecture.

Figure 1: The architecture of SimSiam (Chen and He, 2021) and PhiNets. EMA in the X-PhiNet model stands for the exponential moving average. The architecture originates from a single input, branches out into three paths, and then compares the similarity of all paths in Sim-2. Thus, we call it PhiNet (Φ-Net) because the shape of the architecture resembles the Greek letter Phi (Φ).

The CA3 region receives an input signal from the EC and recurrently forecasts future signals. The prediction output of CA3 is propagated to the CA1 region, which computes the discrepancy between the CA3 prediction and the EC input and refines the internal model stored in CA3. Compensating for the time differences between EC–CA3 and EC–CA1 is hypothesised to facilitate the learning and replay of time sequences in the hippocampus.

Whereas Chen et al. (2024) tested this model to replicate recorded neural activities through simulation, we develop a self-supervised learner PhiNet based on this hypothesis as follows:

- We use deep encoders $f$ and/or $f_{\text{long}}$ to represent cortex. See Section 3.2 for more details.
- We model CA3 by a predictor network.
- We model CA1 by combining a loss function and another predictor.
- We train the model by jointly minimizing the loss for the hippocampus and the neocortex models.
- The long-term memory is implemented by an exponential moving average.

Figures 1a and 1b depict the architecture of Sim-Siam and PhiNet, respectively. Figure 2 illustrates how PhiNet can be interpreted as a hippocampal model (Chen et al., 2024) under the temporal prediction hypothesis.

Note that our approach diverges from the temporal prediction hypothesis method proposed by Chen et al. (2024). Specifically, while they assume an image sequence as input, we consider an original input image and two augmented images as input and feedback signals with time difference (thanks to the StopGradient operation), expanding the applicability of hippocampal models to standard vision tasks.

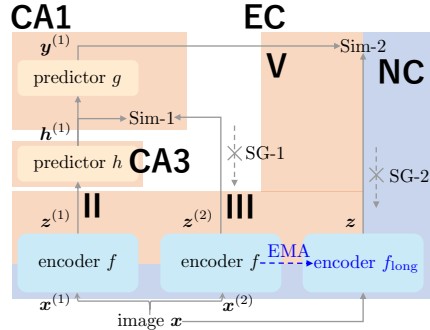

Figure 2: The interpretation as a hippocampal model. NC stands for NeoCortex.

### 3.1 Fast Learning Based on Temporal Prediction Hypothesis

We provide detailed implementation of the hippocampal model, which serves as a fast learner. The model consists of EC, CA3, and CA1, and we describe each of them below.

**Modelling of EC layer.** The entorhinal cortex (EC) is the main input and output cortex of the hippocampus (Chen et al., 2024). Let us denote the original input as $\boldsymbol{x}_i \in \mathbb{R}^d$. We model that the hippocampal model has two augmented signals from the original input as $\boldsymbol{x}_i^{(1)} \in \mathbb{R}^d$ and $\boldsymbol{x}_i^{(2)} \in \mathbb{R}^d$, in addition to the original input $\boldsymbol{x}_i$. Let $f : \mathbb{R}^d \to \mathbb{R}^m$ denote the encoder. Then, the cortical representation in the EC is given as follows:

$$\boldsymbol{z}_i^{(1)} = f(\boldsymbol{x}_i^{(1)}), \quad \boldsymbol{z}_i^{(2)} = f(\boldsymbol{x}_i^{(2)}), \quad \text{and} \quad \boldsymbol{z}_i = f(\boldsymbol{x}_i).$$

We regard each corresponding to the layers II, III, and V of the EC in Chen et al. (2024, Figure 1). For self-supervised training, we have the triplet dataset $\mathcal{D} = \{(\boldsymbol{x}_i^{(1)}, \boldsymbol{x}_i^{(2)}, \boldsymbol{x}_i)\}_{i=1}^n$. The hippocampal learning can be characterized as a learning problem of the encoder $f$ from the training dataset $\mathcal{D}$.

**Modelling of CA3 region.** The CA3 region is responsible for predicting future signals:

$$\boldsymbol{h}_i^{(1)} = h(\boldsymbol{z}_i^{(1)}), \quad \boldsymbol{h}_i^{(2)} = h(\boldsymbol{z}_i^{(2)}),$$

where $h : \mathbb{R}^m \to \mathbb{R}^m$ is the predictor network. We implement the predictor with a two-layer neural network with the ReLU activation and batch normalization.

**Modelling of CA1 region.** CA1 measures the difference between the predicted signal and its future signal. In this paper, we model CA1 by a mixture of a loss function and a predictor, while Chen et al. (2024) uses only the MSE loss for modelling CA1. For the Sim-1 of CA1 depicted in Figure 2, we use the symmetric negative cosine loss function to measure the temporally distant signal $\boldsymbol{z}_i^{(2)}$ (layer III of EC) and the predicted representation from CA3 $\boldsymbol{h}_i^{(1)} = h(\boldsymbol{z}_i^{(1)})$:

$$L_{\mathrm{Cos}}(\boldsymbol{\theta}) = -\frac{1}{2n} \sum_{i=1}^n \frac{(\boldsymbol{h}_i^{(1)})^\top \bar{\boldsymbol{z}}_i^{(2)}}{\|\boldsymbol{h}_i^{(1)}\|_2 \|\bar{\boldsymbol{z}}_i^{(2)}\|_2} - \frac{1}{2n} \sum_{i=1}^n \frac{(\bar{\boldsymbol{z}}_i^{(1)})^\top \boldsymbol{h}_i^{(2)}}{\|\bar{\boldsymbol{z}}_i^{(1)}\|_2 \|\boldsymbol{h}_i^{(2)}\|_2},$$

where $\boldsymbol{\theta}$ represent the entire model parameter and $\bar{\boldsymbol{z}}_i^{(1)} := \mathrm{SG}(\boldsymbol{z}_i^{(1)}) \in \mathbb{R}^m$ is a latent variable with StopGradient, in which the gradient update shall not be executed.

Remark that StopGradient yields a "time difference" during backpropagation, for which we can interpret PhiNet as a hippocampal model. Let us look closely at Sim-1 in Figure 1b. We let $f_t$ denote the encoder $f$ at the $t$-th gradient update. The left path of Sim-1 can then be expressed as $\boldsymbol{h}^{(1)} = h(f_t(\boldsymbol{x}))$. As the right path of Sim-1 is adopted with StopGradient, it can be written as $\boldsymbol{z}^{(2)} = \mathrm{SG}(f_t(\boldsymbol{x})) = f_{t-1}(\boldsymbol{x})$. Eventually, Sim-1 aligns $f_t(\boldsymbol{x})$ and $f_{t-1}(\boldsymbol{x})$ by the predictor $h$. This Sim-1 interpretation indicates that PhiNet predicts *past* signals, which slightly deviates from the original temporal prediction hypothesis (Chen et al., 2024) supposing that CA3 is in charge of predicting *future* signals.

In addition to measuring the difference, the CA1 region outputs the signal to the EC (V layer). Thus, we model the output of CA1 as follows:

$$\boldsymbol{y}_i^{(1)} = g(\boldsymbol{h}_i^{(1)}), \quad \boldsymbol{y}_i^{(2)} = g(\boldsymbol{h}_i^{(2)}),$$

where $g : \mathbb{R}^m \to \mathbb{R}^m$ is another predictor network. As we will see soon, CLS theory supposes that this feedback from CA1 to the EC eventually propagates to the neocortex (NC), which is stored in long-term memory.

### 3.2 INCORPORATING SLOW LEARNING MECHANISM

The hippocampus and neocortex play crucial roles in brain cognition. For effective long-term memory storage, it is essential to transfer information from the hippocampus to the NC. We first aim to formulate the joint learning of the hippocampus and NC models. Then, we propose using the exponential moving average (EMA) to transfer model parameters from short-term to long-term memory, with the goal of compressing the original input signal.

In the EC layer, we model the update of the encoder function by using the output of CA1 and the representation $\boldsymbol{z}_i = f(\boldsymbol{x}_i)$ (V layer of EC). Then, the loss function can be given as follows:

$$L_{\mathrm{NC}}(\boldsymbol{\theta}) = \frac{1}{2n} \sum_{i=1}^n \|\boldsymbol{y}_i^{(1)} - \mathrm{SG}(\boldsymbol{z}_i)\|_2^2 + \frac{1}{2n} \sum_{i=1}^n \|\boldsymbol{y}_i^{(2)} - \mathrm{SG}(\boldsymbol{z}_i)\|_2^2.$$

This corresponds to Sim-2 in Figure 2 and is regarded as slow learning. Finally, the whole objective function of PhiNet is given as

$$L(\boldsymbol{\theta}) = \underbrace{L_{\mathrm{Cos}}(\boldsymbol{\theta})}_{\text{Hippocampus loss / Sim-1}} + \underbrace{L_{\mathrm{NC}}(\boldsymbol{\theta})}_{\text{Neocortex loss / Sim-2}}.$$

We then minimise $L(\boldsymbol{\theta})$ to learn the hippocampus and the NC models. The optimisation can be efficiently performed using backpropagation. It is worth noting that we can utilise different loss functions for Sim-1 and/or Sim-2 in PhiNet. In this paper, we set Sim-1 to negative cosine similarity and Sim-2 to either MSE or the negative cosine similarity.

**X-PhiNet: Slow learning via stable encoder.**   The original PhiNet formulation employs the same encoder for both the short-term and long-term memories for simplicity (Section 3.2). To further enhance slow learning, the input representation in EC-V $z_i$ should maintain long-term signals. Thus, we introduce the following stable encoder for long-term memory:

$$z_i = f_{\text{long}}(x_i).$$

Then, we solve the PhiNet optimisation problem by minimising both $L_{\text{NC}}(\theta)$ and $L_{\text{Cos}}(\theta)$ using the exponential moving average (EMA) of the model parameters of $f$ and $f_{\text{long}}$ as

$$\xi_{\text{long}} \leftarrow \beta \xi_{\text{long}} + (1 - \beta)\xi,$$

where $\xi$ and $\xi_{\text{long}}$ are the model parameters of $f$ and $f_{\text{long}}$, respectively, and $\beta \in [0, 1]$ is a hyperparameter. Model parameters persist in $f_{\text{long}}$ more stably than the original encoder $f$, which facilitates slow learning. We call this method as X-PhiNet.

## 4   WHAT WE BENEFIT FROM ADDITIONAL CA1 PREDICTOR: LEARNING DYNAMICS PERSPECTIVE

When PhiNet is compared with SimSiam, the additional predictor $g$ in CA1 is peculiar. We study the learning dynamics of PhiNet with a toy model. Despite its simplicity, dynamics analysis is beneficial in showcasing how the predictor $g$ effectively prevents complete collapse.

**Analysis model.**   Let us specify the analysis model, following Tian et al. (2021). The $d$-dimensional input is sampled from the isotropic normal $x \sim \mathcal{N}(0, I)$ and augmented by the isotropic normal $x^{(1)}, x^{(2)} \sim \mathcal{N}(x, \sigma^2 I)$, where $\sigma^2$ indicates the strength of data augmentation. The encoder $f$ and predictors $g$ and $h$ are modelled by linear networks without bias: $f(x) := W_f x$, $g(h) := W_g h$, and $h(z) := W_h z$, where $W_f \in \mathbb{R}^{m \times d}$ and $W_g, W_h \in \mathbb{R}^{m \times m}$. The predictors $h$ and $g$ transform latents $z^{(1)}, h^{(1)} \in \mathbb{R}^m$ into $h^{(1)}, y^{(1)} \in \mathbb{R}^m$ with the same dimension $m$ to predict the other noisy latent $z^{(2)}$ and the noise-free latent $z$, respectively (see Figure 1b).

Unlike $L_{\text{Cos}}$ introduced in Section 3, we focus on the (not symmetrised) MSE loss for measuring the discrepancy between $h^{(1)}$ and $z^{(2)}$ for the transparency of analysis. Interested readers may refer to Halvagal et al. (2023) and Bao (2023) for further extension to incorporate the cosine loss into the SimSiam dynamics. Consequently, the expected loss function of PhiNet $\overline{L}(W_f, W_g, W_h)$ is given as follows:

$$\overline{L} := \frac{1}{2}\mathbb{E}_x \mathbb{E}_{x^{(1)}, x^{(2)}|x} \left[ \|W_h W_f x^{(1)} - \text{SG}(W_f x^{(2)})\|^2 + \|W_g W_h W_f x^{(1)} - \text{SG}(W_f x)\|^2 \right].$$

We will analyse the gradient flow $\dot{W}_{\{f,g,h\}} = -\nabla \overline{L} - \rho W_{\{f,g,h\}}$ ($\rho > 0$: weight decay intensity) subsequently. The gradient flows are derived as follows (see Appendix B.1):

$$\dot{W}_f = -W_h^\top \{(1 + \sigma^2)(I + W_g^\top W_g)W_h - (I + W_g^\top)\}W_f - \rho W_f,$$
$$\dot{W}_g = -\{(1 + \sigma^2)W_h - I\}W_f W_f^\top W_h^\top - \rho W_g,$$
$$\dot{W}_h = -\{(1 + \sigma^2)(I + W_g^\top W_g)W_h - (I + W_g^\top)\}W_f W_f^\top - \rho W_h.$$

**Eigenvalue dynamics.**   The matrix dynamics we have derived are rigorous but not amenable to further analysis. Here, we decouple the matrix dynamics into the eigenvalue dynamics. Let $\Phi := W_f W_f^\top \in \mathbb{R}^{m \times m}$. Following Tian et al. (2021, Theorem 3) and Bao (2023, Proposition 1), we can show that the eigenspaces of $\Phi$, $W_g$, and $W_h$ quickly align as $t$ increases (see Appendix B.2). Therefore, we assume the following conditions:

(A1)  $W_g$ and $W_h$ are symmetric.

(A2)  The eigenspaces of $\Phi$, $W_g$, and $W_h$ align for every time step $t$.

Under these assumptions, $\Phi$, $W_g$, and $W_h$ are simultaneously diagonalizable and can be written as $\Phi = U \Lambda_\Phi U^\top$, $W_g = U \Lambda_g U^\top$, and $W_h = U \Lambda_h U^\top$, where $U$ is the (time-dependent) common orthogonal eigenvectors. Here, $\Lambda_\Phi = \text{diag}[\phi_1, \ldots, \phi_m]$, $\Lambda_g = \text{diag}[\gamma_1, \ldots, \gamma_m]$, and

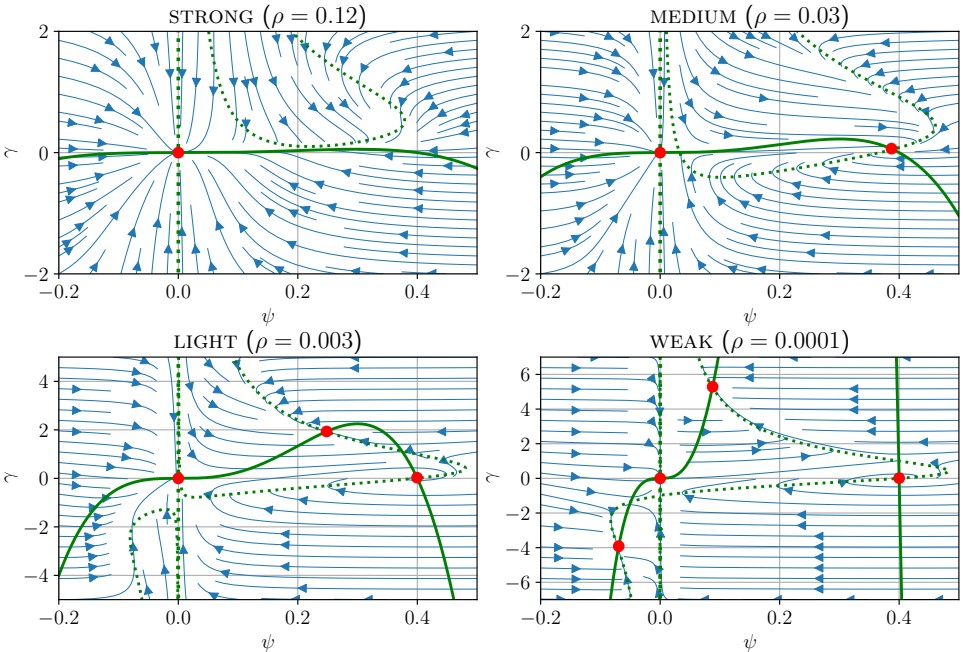

Figure 3: State space diagrams of PhiNet dynamics with different levels of weight decay: STRONG ($\rho = 0.12$), MEDIUM ($\rho = 0.03$), LIGHT ($\rho = 0.003$), and WEAK ($\rho = 0.0001$). The vector fields are numerically computed with $\sigma^2 = 1.5$. The state space bifurcates at the boundary of each level. The nullclines are shown with the green real ($\dot{\psi} = 0$) and dotted ($\dot{\gamma} = 0$) lines. The red dots are sinks.

$\mathbf{\Lambda}_h = \mathrm{diag}[\psi_1, \ldots, \psi_m]$ are the corresponding eigenvalues. Noting that the dynamics quickly falls on to $\phi(t) = \psi(t)^2$, we can decouple the matrix dynamics into the eigenvalue dynamics of $(\psi, \gamma)$ only (shown in Appendix B.3 and B.4):

$$\text{(PhiNet-dynamics)} \quad \begin{cases} \dot{\psi} & = \{(1 + \gamma) - (1 + \sigma^2)(1 + \gamma^2)\psi\}\psi^2 - \rho\psi, \\ \dot{\gamma} & = \{1 - (1 + \sigma^2)\psi\}\psi^3 - \rho\gamma. \end{cases} \tag{1}$$

From the $(\psi, \gamma)$-dynamics, it is easy to see that $(\psi, \gamma) = (0, 0)$ is one of the equilibrium points. Can the eigenvalues escape from this collapsed solution?

**Bifurcation of PhiNet dynamics.** The state space diagrams of dynamics (1) are shown in Figure 3. In this figure, the nullclines $\dot{\psi} = 0$ and $\dot{\gamma} = 0$ are shown in the green real and dotted lines, respectively. Noting that intersecting points of nullclines are equilibrium points (Hirsch et al., 2012), we observe saddle-node bifurcation of PhiNet dynamics parametrized by weight decay $\rho > 0$.

- STRONG: Weight decay $\rho$ is too strong that the collapsed point $(\psi, \gamma) = (0, 0)$ is a unique sink.
- MEDIUM: A new sink $(\psi, \gamma)$ such that $\psi \gg 0$ and $\gamma \approx 0$ emerges. The number of sinks is two.
- LIGHT: Another non-trivial sink $(\psi, \gamma)$ such that $\psi, \gamma \gg 0$ emerges. There are three sinks.
- WEAK: The last sink emerges such that $\psi < 0$ and $\gamma \ll 0$. The number of sinks is four.

**Comparison with SimSiam dynamics.** Tian et al. (2021) derived the SimSiam dynamics under the same setup as above. Specifically, they modelled the encoder $f$ and the predictor $h$ with linear networks $\mathbf{W}_f \boldsymbol{x}$ and $\mathbf{W}_h \boldsymbol{z}$, respectively, and defined the gradient flow dynamics with the MSE loss $\|\mathbf{W}_h \mathbf{W}_f \boldsymbol{x}^{(1)} - \mathrm{SG}(\mathbf{W}_f \boldsymbol{x}^{(2)})\|^2$ (without the additional predictor $g$). By decoupling the matrix dynamics into the eigenvalues with the same adiabatic elimination $\phi = \psi^2$, we can derive the SimSiam dynamics solely with respect to $\psi$-dynamics as follows:

$$\text{(SimSiam-dynamics)} \quad \dot{\psi} = \{1 - (1 + \sigma^2)\psi\}\psi^2 - \rho\psi. \tag{2}$$

We set $\tau = 1$ (ablating the exponential moving average used in BYOL) in Tian et al. (2021, Eq. (16)) to obtain this dynamics. SimSiam is free from the additional predictor $g$, so the dynamics (2) is univariate, unlike the bivariate system (1). Figure 4 shows the dynamics (2).

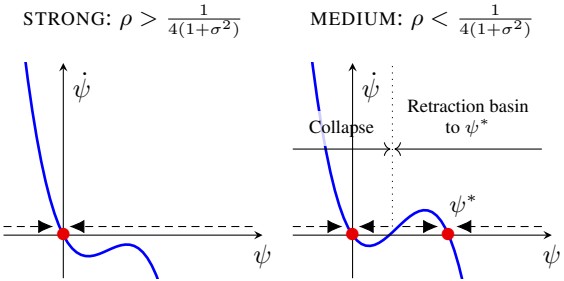

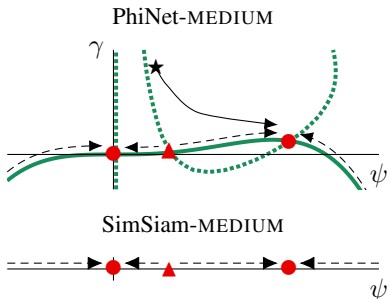

Figure 4: Illustration of SimSiam dynamics (2). Unlike the bivariate PhiNet dynamics shown in Figure 3, SimSiam dynamics is univariate, shown in the $\dot\psi$-axis. The red dots are sinks.

Figure 5: The SimSiam-MEDIUM flow is conjugate with the flow on the nullcline $\dot\psi = 0$ (green real line) in PhiNet-MEDIUM.

Table 1: **PhiNet is comparable to SimSiam.** We trained the models for 100 epochs and then validated them on the test sets using linear probing on the head. We trained with three seeds and calculated means and variances (subscripts). Both are unstable when the weight decay is small, but PhiNet still achieves high accuracy.

| | Accuracy by Linear Probing (w.r.t. weight decay) | | | | | |
| | 0.0 | 0.00001 | 0.00002 | 0.00005 | 0.0001 | 0.0002 |
| --- | --- | --- | --- | --- | --- | --- |
| SimSiam | $25.41_{0.02}$ | $2.63_{0.18}$ | $60.82_{1.57}$ | $44.51_{33.64}$ | $68.17_{0.18}$ | $67.12_{0.13}$ |
| PhiNet (MSE) | $49.89_{0.35}$ | $55.90_{1.57}$ | $33.92_{7.42}$ | $66.73_{0.03}$ | $68.25_{0.21}$ | $67.83_{0.15}$ |

The SimSiam dynamics bifurcates into STRONG and MEDIUM at $\rho = 1/4(1 + \sigma^2)$. These two modes correspond to STRONG and MEDIUM of PhiNet in that $\psi$-axis of Figure 4 and the nullcline $\dot\psi = 0$ (green real line) in Figure 3 are topologically conjugate. The other LIGHT and WEAK are peculiar to the PhiNet dynamics. By comparing Figures 3 and 4, we have the following observations:

(C1) *The retraction basin to non-collapsed solutions is wider*: Since SimSiam dynamics is univariate, $\psi$ cannot avoid collapse once $\psi(0)$ is initialized outside the retraction basin to the non-collapse point $\psi^* \neq 0$ (namely, smaller than the source point ▲ in Figure 5). By contrast, PhiNet avoids collapse even if $\psi(0)$ is close to zero, as long as $\gamma(0)$ is sufficiently large (see the initial point ★ in Figure 5).

(C2) *Even negative initialization $\psi$ can avoid collapse*: In SimSiam-MEDIUM, $\psi$ cannot be attracted to the non-collapsed solution if $\psi(0)$ is initialized to negative. By contrast, PhiNet-WEAK has a negative sink (at the bottom left in Figure 3), which attracts negative initialization $\psi(0) < 0$.

To sum it up, we have witnessed with a toy model that PhiNet is advantageous over SimSiam because the collapsed solution can be avoided more easily. This is why another predictor $g$ is beneficial.

**Remark 1.** *The learning dynamics analysis in this section reveals that smaller weight decay $\rho$ brings us benefits only regarding the stability of non-collapsed solutions. Indeed, we may benefit from larger $\rho$ to accelerate convergence to the invariant parabola and eigenspace alignment of $(\mathbf{\Phi}, \mathbf{W}_h, \mathbf{W}_g)$ (Appendices B.4 and B.2), each of which corresponds to the positive effects #3 and #7 in Tian et al. (2021), respectively. Moreover, moderately large $\rho$ often yields good generalization in non-contrastive learning (Cabannes et al., 2023). Thus, smaller $\rho$ may not be a silver bullet.*

## 5 EXPERIMENTS

We first test the robustness of PhiNets against the design choice and weight decay hyperparameter. We then discuss the effectiveness of X-PhiNets in online and continual learning.

### 5.1 LINEAR PROBING ANALYSIS

Figure 6 and Table 1 show the sensitivity analysis using CIFAR10 (Krizhevsky, 2009) and ImageNet (Krizhevsky et al., 2012), respectively, by changing the weight decay parameter. First, we

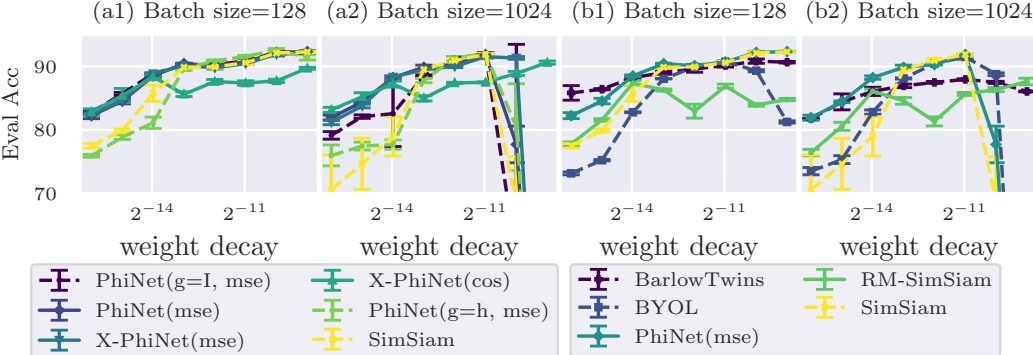

Figure 6: **PhiNet and X-PhiNet are robust against weight decay.** We evaluated PhiNet variants in (a1)-(a2) and compared the existing non-contrastive methods with PhiNet in (b1)-(b2) on CIFAR10. We evaluated the performance using linear probing. The loss function in brackets represents neocortex loss $L_{\text{NC}}(\boldsymbol{\theta})$. PhiNets perform particularly better than the baselines when weight decay is small.

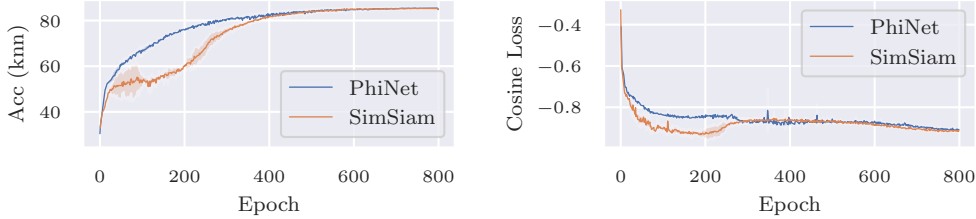

Figure 7: **PhiNet is stable in the early stages of learning.** We trained PhiNet and SimSiam with a batch size of $1024$ and the weight decay of $1e-4$ on STL10. SimSiam is unstable in the early stages of learning. This may be due to the cosine loss being too small in SimSiam.

emphasise the improvement of PhiNet over SimSiam for most of the setups, supporting the importance of the CA1 predictor and Sim-2 loss. Subsequently, we closely look at the results.

**PhiNet improves SimSiam.** We observed weight decay significantly impacts the final model performance. When the MSE loss is used for Sim-2, PhiNet consistently outperforms the original SimSiam or other baselines (BYOL, RM-SimSiam) regardless of weight decay value, shown in Figure 6 (right). Moreover, as shown in Figure 7, PhiNet have a stabilizing effect during the early stages of training. This can likely be attributed to PhiNet's regularization effect, which prevents the cosine loss from becoming too small at the early phase of training.

**Bless of additional CA1 predictor.** To see whether the additional predictor $g$ besides $h$ is beneficial, we test variants of predictor $g$: $g = h$ (reminiscent of the recurrent structure in CA3) and $g = \mathbf{I}$ (identity predictor). For CIFAR10 with batch size = 128, Figure 6 (left) indicates that the predictor choice slightly affects the final model performance if we properly set the weight decay. However, if we set the batch size as 1024, the separate predictor performs more stably over other choices.

**Sim-2 loss should be MSE.** Based on Figure 6 and Table 11 in the appendix, we found that the MSE loss used for Sim-2 generally improves model performance across most weight decay parameters, while the negative cosine loss performs comparably to the MSE loss with smaller weight decay but degrades it with larger weight decay.

Overall, our sensitivity study on CIFAR10 revealed that PhiNets are robust to the choice of the weight decay parameter, which supports the importance of the CA1 predictor and Sim-2 loss. See Appendix E.1 for more detailed sensitivity studies. In addition, the results for different batch sizes and datasets (STL10 (Coates et al., 2011)) can be found in Appendix E.4.

## 5.2 ONLINE LEARNING AND CONTINUAL LEARNING

SimSiam and other non-contrastive methods typically require up to 800 epochs of training on CIFAR10, which is quite different from the online nature of brains. To address this, we conducted

Table 2: **X-PhiNet performs good results when memorization is important.** We trained PhiNets on CIFAR-5m and Split CIFAR-5m. In Split CIFAR-5m, Acc is the average of the final accuracy (higher is better), and Fg is Forgetting (smaller is better). We present the results for two different weight decay (5e − 4 and 2e − 5) in Split CIFAR-5m.

| | | BYOL | SimSiam | Barlow Twins | PhiNet (MSE) | RM-SimSiam | X-PhiNet (MSE) | X-PhiNet (Cos) |
|---|---|---|---|---|---|---|---|---|
| CIFAR-5m | | $81.05_{0.04}$ | $77.71_{1.97}$ | $85.32_{0.10}$ | $76.74_{1.82}$ | $82.09_{0.22}$ | $87.30_{0.13}$ | $\mathbf{87.46_{0.14}}$ |
| Split C-5m | Acc | $90.44_{0.28}$ | $90.84_{0.31}$ | $90.30_{0.17}$ | $90.69_{0.11}$ | $90.04_{0.11}$ | $91.02_{0.36}$ | $\mathbf{92.83_{0.12}}$ |
| (wd=5e-4) | Fg | $1.61_{0.50}$ | $2.45_{0.42}$ | $3.36_{1.10}$ | $2.96_{0.23}$ | $2.44_{0.22}$ | $3.44_{0.36}$ | $1.95_{0.19}$ |
| Split C-5m | Acc | $86.87_{0.12}$ | $88.20_{0.35}$ | $89.86_{0.34}$ | $88.60_{0.15}$ | $87.07_{0.36}$ | $\mathbf{90.90_{0.38}}$ | $90.72_{0.23}$ |
| (wd=2e-5) | Fg | $-0.16_{0.23}$ | $0.36_{0.51}$ | $1.29_{0.94}$ | $0.05_{0.19}$ | $0.82_{0.14}$ | $-1.03_{0.41}$ | $0.43_{0.17}$ |

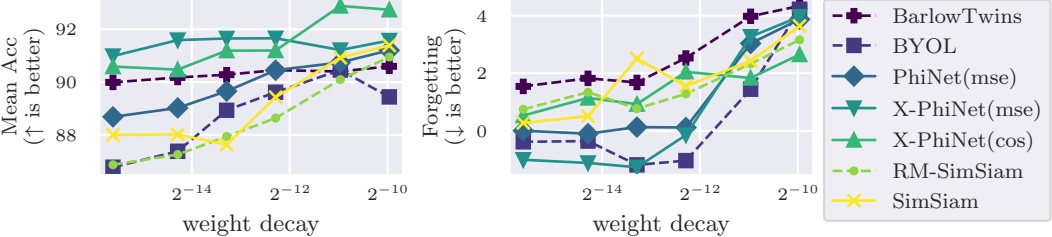

Figure 8: **X-PhiNet is also robust to weight decay in continual learning.** We measured the mean accuracy and forgetting at different weight decay on Split CIFAR-5m.

experiments using the CIFAR-5m dataset, which has six million synthetic CIFAR10-like images generated by the DDPM generative model (Nakkiran et al., 2021). Instead of training CIFAR10 with 50k samples for 800 epochs, we trained CIFAR-5m with 5m samples for 8 epochs. Although this is not exactly online learning, it seems closer to online learning compared to CIFAR10 due to the restriction on the training epochs. Table 2 shows that X-PhiNet has higher accuracy than SimSiam and PhiNet. The superior performance of X-PhiNet compared to PhiNet suggests that long-term memory with EMA is important in online learning. Sensitivity to weight decay and results for one-epoch online learning are given in Appendix D.1.

X-PhiNet draws inspiration from CLS theory, which proposes a framework for understanding continual learning processes in human brains. To evaluate the effectiveness of X-PhiNet in continual learning, we created a split CIFAR-5m dataset from CIFAR-5m, dividing it into five tasks, each with two classes. We trained on each task for one epoch and evaluated performance by the average accuracy across all tasks and the average forgetting, which is the difference between the peak accuracy and the final accuracy of each task. Table 2 shows that X-PhiNet has higher performance than SimSiam while maintaining minimal forgetting. Figure 8 further demonstrates that X-PhiNet consistently outperforms other methods like SimSiam in continual learning, regardless of weight decay. X-PhiNet also demonstrates high performance on Split CIFAR10 and Split CIFAR100, as well as when using replay methods (Appendix D.2).

## 6 CONCLUSION

In this paper, we proposed PhiNets based on non-contrastive learning with the temporal prediction hypothesis. Specifically, we leveraged StopGradient to artificially simulate the synaptic delay, and the prediction errors are modelled via Sim-1 and Sim-2 losses. Through theoretical analysis of learning dynamics, we showed that the proposed PhiNets have an advantage over SimSiam by more easily avoiding collapsed solutions. We empirically validated that the proposed PhiNets are robust with respect to weight decay and favorably comparable with SimSiam in terms of final classification performance. Experimental results also show that X-PhiNet performs better than SimSiam in online and continual learning, where memory function matters. These findings corroborate the effectiveness of the temporal prediction hypothesis when robustness and adaptivity are important.

ACKNOWLEDGEMENT

The authors thank Kenji Doya and Tomoki Fukaki for their helpful comments. We also thank Rio Yokota for providing computational resources. M.Y. and H.B. were supported by JSPS KAKENHI Grant Number 24K03004, H.B. was supported by JST PRESTO Grant Number JPMJPR24K6, and Y.T. was supported by JSPS KAKENHI Grant Number 23KJ1336.

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

## A   LIMITATIONS AND FUTURE WORK (EXTENDED VERSION)

One major limitation of our approach is the use of backpropagation, which differs from the mechanisms in biological neural networks. Our long-term goal is to eliminate backpropagation to better imitate brain function, but this work focuses on the model's structural aspects. Currently, backpropagation-free predictive coding mechanisms for complex architectures like ResNet are in the early stages of development, with most research limited to simple CNNs. Conversely, non-contrastive methods like SimSiam require more advanced models than ResNet. Future research should explore if the proposed structure can enable effective learning with backpropagation-free predictive coding. Another key difference between PhiNet and brains is the presence of recurrent structures. However, in this PhiNet, only one time step is considered, so it is possible that the recurrent structure required to predict time series data was not necessary. Making the data into time series data and adding a recurrent structure to the model remains as future work.

It is also unclear whether cosine loss or MSE loss is more suitable for the Sim-2 in PhiNets. Cosine loss performs better when weight decay is small or online and continual learning where the additional predictor of PhiNets is important. However, MSE loss is preferable when weight decay is large on CIFAR10. This is likely because using cosine loss in sim-2 has a stronger impact on learning dynamics compared to MSE loss. Analyzing gradient norms could be useful for this kind of evaluation, but is left for future work.

## B   DETAILS OF LEARNING DYNAMICS ANALYSIS

In this section, we complement the missing details of learning dynamics analysis provided in Section 4. In our analysis, we will use the gradient flow $\dot{\mathbf{W}}_{\{f,g,h\}} = -\nabla\overline{L} - \rho\mathbf{W}_{\{f,g,h\}}$ ($\rho > 0$), which is the continuous limit of gradient descent. This corresponds to considering the following gradient descent in discrete updates and taking the limit as $\eta \to 0$.

$$\mathbf{W}_{\{f,g,h\}}(t+\eta) = \mathbf{W}_{\{f,g,h\}}(t) - \eta\nabla\overline{L} - \eta\rho\mathbf{W}_{\{f,g,h\}} \tag{3}$$

$$\frac{1}{\eta}\left(\mathbf{W}_{\{f,g,h\}}(t+\eta) - \mathbf{W}_{\{f,g,h\}}(t)\right) = -\nabla\overline{L} - \rho\mathbf{W}_{\{f,g,h\}} \tag{4}$$

### B.1   DERIVATION OF MATRIX DYNAMICS

Recall the PhiNet loss function:

$$\overline{L} := \frac{1}{2}\mathbb{E}_{\boldsymbol{x}}\mathbb{E}_{\boldsymbol{x}^{(1)},\boldsymbol{x}^{(2)}|\boldsymbol{x}}\left[\|\mathbf{W}_h\mathbf{W}_f\boldsymbol{x}^{(1)} - \text{SG}(\mathbf{W}_f\boldsymbol{x}^{(2)})\|^2 + \|\mathbf{W}_g\mathbf{W}_h\mathbf{W}_f\boldsymbol{x}^{(1)} - \text{SG}(\mathbf{W}_f\boldsymbol{x})\|^2\right].$$

Let us derive its matrix gradient.

$$\nabla_{\mathbf{W}_f}\overline{L} = \frac{1}{2}\nabla_{\mathbf{W}_f}\mathbb{E}\left[(\boldsymbol{x}^{(1)\top}\mathbf{W}_f^\top\mathbf{W}_h^\top - \text{SG}(\boldsymbol{x}^{(2)\top}\mathbf{W}_f^\top))(\mathbf{W}_h\mathbf{W}_f\boldsymbol{x}^{(1)} - \text{SG}(\mathbf{W}_f\boldsymbol{x}^{(2)}))\right.$$
$$\left. + (\boldsymbol{x}^{(1)}\mathbf{W}_f^\top\mathbf{W}_h^\top\mathbf{W}_g^\top - \text{SG}(\boldsymbol{x}^\top\mathbf{W}_f^\top))(\mathbf{W}_g\mathbf{W}_h\mathbf{W}_f\boldsymbol{x}^{(1)} - \text{SG}(\mathbf{W}_f\boldsymbol{x}))\right]$$
$$= \left\{\mathbf{W}_h^\top\mathbf{W}_h\mathbf{W}_f\mathbb{E}[\boldsymbol{x}^{(1)}\boldsymbol{x}^{(1)\top}] - \mathbf{W}_h^\top\mathbf{W}_f\mathbb{E}[\boldsymbol{x}^{(2)}\boldsymbol{x}^{(1)\top}]\right\}$$
$$+ \left\{\mathbf{W}_h^\top\mathbf{W}_g^\top\mathbf{W}_g\mathbf{W}_h\mathbf{W}_f\mathbb{E}[\boldsymbol{x}^{(1)}\boldsymbol{x}^{(1)\top}] - \mathbf{W}_h^\top\mathbf{W}_g^\top\mathbf{W}_f\mathbb{E}[\boldsymbol{x}\boldsymbol{x}^{(1)\top}]\right\}$$
$$= \mathbf{W}_h^\top\left\{(\mathbf{I} + \mathbf{W}_g^\top\mathbf{W}_g)\mathbf{W}_h\mathbf{W}_f\mathbb{E}[\boldsymbol{x}^{(1)}\boldsymbol{x}^{(1)\top}] - \mathbf{W}_f\mathbb{E}[\boldsymbol{x}^{(2)}\boldsymbol{x}^{(1)\top}] - \mathbf{W}_g^\top\mathbf{W}_f\mathbb{E}[\boldsymbol{x}\boldsymbol{x}^{(1)\top}]\right\}$$
$$= \mathbf{W}_h^\top\left\{(1 + \sigma^2)(\mathbf{I} + \mathbf{W}_g^\top\mathbf{W}_g)\mathbf{W}_h - (\mathbf{I} + \mathbf{W}_g^\top)\right\}\mathbf{W}_f,$$

where the last line is derived from our assumption on the data distributions:

$$\mathbb{E}_{\boldsymbol{x}}\mathbb{E}_{\boldsymbol{x}^{(1)}|\boldsymbol{x}}[\boldsymbol{x}^{(1)}\boldsymbol{x}^{(1)\top}] = \mathbb{E}_{\boldsymbol{x}}[\boldsymbol{x}\boldsymbol{x}^\top] + \sigma^2\mathbf{I} = (1 + \sigma^2)\mathbf{I},$$
$$\mathbb{E}_{\boldsymbol{x}}\mathbb{E}_{\boldsymbol{x}^{(1)},\boldsymbol{x}^{(2)}|\boldsymbol{x}}[\boldsymbol{x}^{(2)}\boldsymbol{x}^{(1)\top}] = \mathbb{E}_{\boldsymbol{x}}[\boldsymbol{x}\boldsymbol{x}^\top] = \mathbf{I},$$
$$\mathbb{E}_{\boldsymbol{x}}\mathbb{E}_{\boldsymbol{x}^{(1)}|\boldsymbol{x}}[\boldsymbol{x}\boldsymbol{x}^{(1)\top}] = \mathbb{E}_{\boldsymbol{x}}[\boldsymbol{x}\boldsymbol{x}^\top] = \mathbf{I}.$$

Similarly, we derive $\nabla_{\mathbf{W}_g}\overline{L}$ and $\nabla_{\mathbf{W}_h}\overline{L}$.

$$\nabla_{\mathbf{W}_g}\overline{L} = \mathbf{W}_h\mathbf{W}_f\mathbb{E}[\boldsymbol{x}^{(1)}\boldsymbol{x}^{(1)\top}]\mathbf{W}_f^\top\mathbf{W}_h^\top - \mathbf{W}_f\mathbb{E}[\boldsymbol{x}\boldsymbol{x}^{(1)\top}]\mathbf{W}_f^\top\mathbf{W}_h^\top$$
$$= \left\{(1+\sigma^2)\mathbf{W}_h - \mathbf{I}\right\}\mathbf{W}_f\mathbf{W}_f^\top\mathbf{W}_h^\top.$$

$$\nabla_{\mathbf{W}_h}\overline{L} = \left\{\mathbf{W}_h\mathbf{W}_f\mathbb{E}[\boldsymbol{x}^{(1)}\boldsymbol{x}^{(1)\top}]\mathbf{W}_f^\top - \mathbf{W}_f\mathbb{E}[\boldsymbol{x}^{(2)}\boldsymbol{x}^{(1)\top}]\mathbf{W}_f^\top\right\}$$
$$+ \left\{\mathbf{W}_g^\top\mathbf{W}_g\mathbf{W}_h\mathbf{W}_f\mathbb{E}[\boldsymbol{x}^{(1)}\boldsymbol{x}^{(1)\top}]\mathbf{W}_f^\top - \mathbf{W}_g^\top\mathbf{W}_f\mathbb{E}[\boldsymbol{x}\boldsymbol{x}^{(1)\top}]\mathbf{W}_f^\top\right\}$$
$$= \left\{(1+\sigma^2)(\mathbf{I}+\mathbf{W}_g^\top\mathbf{W}_g)\mathbf{W}_h - (\mathbf{I}+\mathbf{W}_g^\top)\right\}\mathbf{W}_f\mathbf{W}_f^\top.$$

From these, we obtain the matrix dynamics.

## B.2 EIGENSPACE ALIGNMENT

Our aim is to show that the three matrices $\boldsymbol{\Phi}$, $\mathbf{W}_g$, and $\mathbf{W}_h$ share a common eigenspace, i.e., simultaneously diagonalizable, asymptotically in time $t$. Let

$$\mathbf{C}_1 \coloneqq [\boldsymbol{\Phi}, \mathbf{W}_g], \quad \mathbf{C}_2 \coloneqq [\boldsymbol{\Phi}, \mathbf{W}_h], \quad \text{and} \quad \mathbf{C}_3 \coloneqq [\mathbf{W}_g, \mathbf{W}_h],$$

where $[\mathbf{A}, \mathbf{B}] \coloneqq \mathbf{A}\mathbf{B} - \mathbf{B}\mathbf{A}$ is the commutator (matrix). By noting that commutative matrices are simultaneously diagonalizable, we show that the time-dependent commutators $\mathbf{C}_1(t)$, $\mathbf{C}_2(t)$, and $\mathbf{C}_3(t)$ asymptotically converges to $\mathbf{O}$ as $t \to \infty$.

Hereafter, we assume the symmetry assumption (A1) on $\mathbf{W}_g$ and $\mathbf{W}_h$, and heavily use the following formulas on commutators implicitly:

- $[\mathbf{A}, \mathbf{A}] = \mathbf{O}$.
- $[\mathbf{A}, \mathbf{B}] = -[\mathbf{B}, \mathbf{A}]$.
- $[\mathbf{A}, \mathbf{BC}] = [\mathbf{A}, \mathbf{B}]\mathbf{C} + \mathbf{B}[\mathbf{A}, \mathbf{C}]$.
- $[\mathbf{AB}, \mathbf{C}] = \mathbf{A}[\mathbf{B}, \mathbf{C}] + [\mathbf{A}, \mathbf{C}]\mathbf{B}$.

First, compute $\dot{\mathbf{C}}_1$ based on the matrix dynamics of $\boldsymbol{\Phi}$ (can be found in Appendix B.3) and $\mathbf{W}_g$:

$$\dot{\mathbf{C}}_1 = \dot{\boldsymbol{\Phi}}\mathbf{W}_g + \boldsymbol{\Phi}\dot{\mathbf{W}}_g - \dot{\mathbf{W}}_g\boldsymbol{\Phi} - \mathbf{W}_g\dot{\boldsymbol{\Phi}}$$
$$= \{-(1+\sigma^2)(\mathbf{W}_h(\mathbf{I}+\mathbf{W}_g^2)\mathbf{W}_h\boldsymbol{\Phi} + \boldsymbol{\Phi}\mathbf{W}_h(\mathbf{I}+\mathbf{W}_g^2)\mathbf{W}_h) + (\mathbf{W}_h\boldsymbol{\Phi} + \boldsymbol{\Phi}\mathbf{W}_h)$$
$$\quad + (\mathbf{W}_h\mathbf{W}_g\boldsymbol{\Phi} + \boldsymbol{\Phi}\mathbf{W}_g\mathbf{W}_h) - 2\rho\boldsymbol{\Phi}\}\mathbf{W}_g + \boldsymbol{\Phi}\{-((1+\sigma^2)\mathbf{W}_h - \mathbf{I})\boldsymbol{\Phi}\mathbf{W}_h - \rho\mathbf{W}_g\}$$
$$\quad + \{((1+\sigma^2)\mathbf{W}_h - \mathbf{I})\boldsymbol{\Phi}\mathbf{W}_h + \rho\mathbf{W}_g\}\boldsymbol{\Phi}$$
$$\quad + \mathbf{W}_g\{(1+\sigma^2)(\mathbf{W}_h(\mathbf{I}+\mathbf{W}_g^2)\mathbf{W}_h\boldsymbol{\Phi} + \boldsymbol{\Phi}\mathbf{W}_h(\mathbf{I}+\mathbf{W}_g^2)\mathbf{W}_h) - (\mathbf{W}_h\boldsymbol{\Phi} + \boldsymbol{\Phi}\mathbf{W}_h)$$
$$\quad - (\mathbf{W}_h\mathbf{W}_g\boldsymbol{\Phi} + \boldsymbol{\Phi}\mathbf{W}_g\mathbf{W}_h) + 2\rho\boldsymbol{\Phi}\}$$
$$= -3\rho\mathbf{C}_1 + (1+\sigma^2)[\mathbf{W}_g, \mathbf{W}_h(\mathbf{I}+\mathbf{W}_g^2)\mathbf{W}_h\boldsymbol{\Phi}] + (1+\sigma^2)[\mathbf{W}_g, \boldsymbol{\Phi}\mathbf{W}_h(\mathbf{I}+\mathbf{W}_g^2)\mathbf{W}_h]$$
$$\quad + [\mathbf{W}_h\boldsymbol{\Phi} + \boldsymbol{\Phi}\mathbf{W}_h, \mathbf{W}_g] + [\mathbf{W}_h, \mathbf{W}_g\boldsymbol{\Phi}\mathbf{W}_g] + [\boldsymbol{\Phi}, \mathbf{W}_g\mathbf{W}_h\mathbf{W}_g]$$
$$\quad + [((1+\sigma^2)\mathbf{W}_h - \mathbf{I})\boldsymbol{\Phi}\mathbf{W}_h, \boldsymbol{\Phi}]$$
$$= -3\rho\mathbf{C}_1 + (1+\sigma^2)\{(\mathbf{C}_3\mathbf{W}_h\boldsymbol{\Phi} + \boldsymbol{\Phi}\mathbf{W}_h\mathbf{C}_3) + (\mathbf{W}_h\mathbf{C}_3\boldsymbol{\Phi} + \boldsymbol{\Phi}\mathbf{C}_3\mathbf{W}_h)$$
$$\quad + (\mathbf{C}_3\mathbf{W}_g^2\mathbf{W}_h + \mathbf{W}_h\mathbf{W}_g^2\mathbf{C}_3)\boldsymbol{\Phi} - (\mathbf{W}_h\mathbf{W}_g^2\mathbf{W}_h\mathbf{C}_1 + \mathbf{C}_1\mathbf{W}_h\mathbf{W}_g^2\mathbf{W}_h)$$
$$\quad - (\mathbf{W}_h^2\mathbf{C}_1 + \mathbf{C}_1\mathbf{W}_h^2) + \boldsymbol{\Phi}(\mathbf{C}_3\mathbf{W}_g^2\mathbf{W}_h + \mathbf{W}_h\mathbf{W}_g^2\mathbf{C}_3)\}$$
$$\quad + (\mathbf{W}_h\mathbf{C}_1 + \mathbf{C}_1\mathbf{W}_h) - (\mathbf{C}_3\boldsymbol{\Phi} + \boldsymbol{\Phi}\mathbf{C}_3) - (\mathbf{C}_3\boldsymbol{\Phi}\mathbf{W}_g + \mathbf{W}_g\boldsymbol{\Phi}\mathbf{C}_3)$$
$$\quad + (\mathbf{C}_1\mathbf{W}_h\mathbf{W}_g + \mathbf{W}_g\mathbf{W}_h\mathbf{C}_1) - (1+\sigma^2)(\mathbf{W}_h\boldsymbol{\Phi}\mathbf{C}_2 + \mathbf{C}_2\boldsymbol{\Phi}\mathbf{W}_h) + \boldsymbol{\Phi}\mathbf{C}_2.$$

Similarly, $\dot{\mathbf{C}}_2$ and $\dot{\mathbf{C}}_3$ are computed:

$$
\begin{aligned}
\dot{\mathbf{C}}_2 &= -3\rho\mathbf{C}_2 + (\mathbf{C}_2\mathbf{W}_h + \mathbf{W}_h\mathbf{C}_2) - \mathbf{C}_1\boldsymbol{\Phi} + (\mathbf{W}_g\mathbf{C}_2 + \mathbf{C}_2\mathbf{W}_g) + (\mathbf{C}_3\boldsymbol{\Phi} + \boldsymbol{\Phi}\mathbf{C}_3) \\
&\quad - (1+\sigma^2)\mathbf{C}_2\boldsymbol{\Phi} - (1+\sigma^2)(\mathbf{W}_g\mathbf{C}_1 + \mathbf{C}_1\mathbf{W}_g) + (1+\sigma^2)\mathbf{W}_h(\mathbf{C}_3\mathbf{W}_g + \mathbf{W}_g\mathbf{C}_3) \\
&\quad - (1+\sigma^2)(\mathbf{W}_h(\mathbf{I}+\mathbf{W}_g^2)\mathbf{W}_h\mathbf{C}_2 + \mathbf{C}_2\mathbf{W}_h(\mathbf{I}+\mathbf{W}_g^2)\mathbf{W}_h) \\
&\quad + (1+\sigma^2)\boldsymbol{\Phi}\mathbf{W}_h(\mathbf{C}_3\mathbf{W}_g + \mathbf{W}_g\mathbf{C}_3)\mathbf{W}_h, \\
\dot{\mathbf{C}}_3 &= -3\rho\mathbf{C}_3 + (\mathbf{I} - (1+\sigma^2)\mathbf{W}_h)\mathbf{C}_2\mathbf{W}_h + (1+\sigma^2)(\mathbf{I}+\mathbf{W}_g^2)(\mathbf{W}_h\mathbf{C}_1 - \mathbf{C}_3\boldsymbol{\Phi}) \\
&\quad + (\mathbf{I} + \mathbf{W}_g)\mathbf{C}_1.
\end{aligned}
$$

Next, we vectorize the commutator matrices—for $\mathbf{C} \in \mathbb{R}^{h \times h}$, $\operatorname{vec}(\mathbf{C}) \in \mathbb{R}^{h^2}$ indicates a (column) vector stacking the columns of $\mathbf{C}$. For the commutators $\mathbf{C}_1$, $\mathbf{C}_2$, and $\mathbf{C}_3$, let us write $\boldsymbol{\xi} := \operatorname{vec}(\mathbf{C}_1)$, $\boldsymbol{\eta} := \operatorname{vec}(\mathbf{C}_2)$, and $\boldsymbol{\zeta} := \operatorname{vec}(\mathbf{C}_3)$. In what follows, we heavily leverage the vectorization formula:

- $\operatorname{vec}(\mathbf{ABC}) = (\mathbf{C}^\top \otimes \mathbf{A})\operatorname{vec}(\mathbf{B}) = (\mathbf{I} \otimes \mathbf{AB})\operatorname{vec}(\mathbf{C}) = (\mathbf{C}^\top\mathbf{B}^\top \otimes \mathbf{I})\operatorname{vec}(\mathbf{A})$
- $\operatorname{vec}(\mathbf{AB}) = (\mathbf{I} \otimes \mathbf{A})\operatorname{vec}(\mathbf{B}) = (\mathbf{B}^\top \otimes \mathbf{I})\operatorname{vec}(\mathbf{A})$

Here, $\mathbf{A} \otimes \mathbf{B}$ denotes the Kronecker product of two matrices. We write $\mathbf{A} \oplus \mathbf{B} := \mathbf{A} \otimes \mathbf{B} + \mathbf{B} \otimes \mathbf{A}$ for notational convenience. We derive the ODE of $\boldsymbol{\xi} = \operatorname{vec}(\mathbf{C}_1)$ as follows:

$$
\begin{aligned}
\dot{\boldsymbol{\xi}} &= -3\rho\mathbf{I}\boldsymbol{\xi} + (1+\sigma^2)((\boldsymbol{\Phi}\mathbf{W}_h \oplus \mathbf{I})\boldsymbol{\zeta} + (\boldsymbol{\Phi} \oplus \mathbf{W}_h)\boldsymbol{\zeta} + (\boldsymbol{\Phi} \otimes \mathbf{I})(\mathbf{W}_h\mathbf{W}_g^2 \oplus \mathbf{I})\boldsymbol{\zeta} \\
&\quad - (\mathbf{W}_h\mathbf{W}_g^2\mathbf{W}_h \oplus \mathbf{I})\boldsymbol{\xi} - (\mathbf{I} \oplus \mathbf{W}_h^2)\boldsymbol{\xi} + (\mathbf{I} \otimes \boldsymbol{\Phi})(\mathbf{W}_h\mathbf{W}_g^2 \oplus \mathbf{I})\boldsymbol{\zeta}) + (\mathbf{I} \oplus \mathbf{W}_h)\boldsymbol{\xi} \\
&\quad - (\boldsymbol{\Phi} \oplus \mathbf{I})\boldsymbol{\zeta} - (\mathbf{W}_g\boldsymbol{\Phi} \oplus \mathbf{I})\boldsymbol{\zeta} + (\mathbf{W}_g\mathbf{W}_h \oplus \mathbf{I})\boldsymbol{\xi} - (1+\sigma^2)(\mathbf{I} \oplus \mathbf{W}_h\boldsymbol{\Phi})\boldsymbol{\eta} + (\mathbf{I} \otimes \boldsymbol{\Phi})\boldsymbol{\eta} \\
&= -\{3\rho\mathbf{I} + \mathbf{I} \oplus ((1+\sigma^2)\mathbf{W}_h(\mathbf{I}+\mathbf{W}_g^2)\mathbf{W}_h) + \mathbf{W}_h(\mathbf{I}+\mathbf{W}_g)\}\boldsymbol{\xi} \\
&\quad - \{(1+\sigma^2)(\mathbf{I} \otimes \mathbf{W}_h\boldsymbol{\Phi}) - \mathbf{I} \otimes \boldsymbol{\Phi}\}\boldsymbol{\eta} \\
&\quad - \{(\mathbf{I} + \mathbf{W}_g)\boldsymbol{\Phi} \oplus \mathbf{I} - (1+\sigma^2)(\boldsymbol{\Phi}\mathbf{W}_h \oplus \mathbf{I} + (\mathbf{I} \oplus \boldsymbol{\Phi})(\mathbf{W}_h\mathbf{W}_g^2 \oplus \mathbf{I}))\}\boldsymbol{\zeta} \\
&= -(3\rho\mathbf{I} + \mathbf{K}_{11})\boldsymbol{\xi} - \mathbf{K}_{12}\boldsymbol{\eta} - \mathbf{K}_{13}\boldsymbol{\zeta},
\end{aligned}
$$

where

$$
\begin{aligned}
\mathbf{K}_{11} &= \mathbf{I} \oplus ((1+\sigma^2)\mathbf{W}_h(\mathbf{I}+\mathbf{W}_g^2)\mathbf{W}_h) + \mathbf{W}_h(\mathbf{I}+\mathbf{W}_g), \\
\mathbf{K}_{12} &= (1+\sigma^2)(\mathbf{I} \otimes \mathbf{W}_h\boldsymbol{\Phi}) - \mathbf{I} \otimes \boldsymbol{\Phi}, \\
\mathbf{K}_{13} &= (\mathbf{I} + \mathbf{W}_g)\boldsymbol{\Phi} \oplus \mathbf{I} - (1+\sigma^2)(\boldsymbol{\Phi}\mathbf{W}_h \oplus \mathbf{I} + (\mathbf{I} \oplus \boldsymbol{\Phi})(\mathbf{W}_h\mathbf{W}_g^2 \oplus \mathbf{I})).
\end{aligned}
$$

Similarly, we derive the ODEs of $\boldsymbol{\eta} = \operatorname{vec}(\mathbf{C}_2)$ and $\boldsymbol{\zeta} = \operatorname{vec}(\mathbf{C}_3)$.

$$
\begin{aligned}
\dot{\boldsymbol{\eta}} &= -\mathbf{K}_{21}\boldsymbol{\xi} - (3\rho\mathbf{I} + \mathbf{K}_{22})\boldsymbol{\eta} - \mathbf{K}_{23}\boldsymbol{\zeta}, \\
\dot{\boldsymbol{\zeta}} &= -\mathbf{K}_{31}\boldsymbol{\xi} - \mathbf{K}_{32}\boldsymbol{\eta} - (3\rho\mathbf{I} + \mathbf{K}_{33})\boldsymbol{\zeta},
\end{aligned}
$$

where

$$
\begin{aligned}
\mathbf{K}_{21} &= \boldsymbol{\Phi} \otimes \mathbf{I} + (1+\sigma^2)(\mathbf{W}_g \oplus \mathbf{I}), \\
\mathbf{K}_{22} &= (1+\sigma^2)(\boldsymbol{\Phi} \otimes \mathbf{I}) + \mathbf{I} \oplus \{(1+\sigma^2)(\mathbf{W}_h(\mathbf{I}+\mathbf{W}_g^2)\mathbf{W}_h - (\mathbf{W}_g + \mathbf{W}_h))\}, \\
\mathbf{K}_{23} &= -\mathbf{I} \oplus \boldsymbol{\Phi} - (1+\sigma^2)\{(\mathbf{I} \otimes \mathbf{W}_h) + (\mathbf{W}_h \otimes \boldsymbol{\Phi}\mathbf{W}_h)\}(\mathbf{I} \oplus \mathbf{W}_g), \\
\mathbf{K}_{31} &= -(1+\sigma^2)(\mathbf{I} \otimes (\mathbf{I}+\mathbf{W}_g^2)\mathbf{W}_h) - \mathbf{I} \otimes (\mathbf{I}+\mathbf{W}_g), \\
\mathbf{K}_{32} &= -\mathbf{W}_h \otimes (\mathbf{I} - (1+\sigma^2)\mathbf{W}_h), \\
\mathbf{K}_{33} &= (1+\sigma^2)(\boldsymbol{\Phi} \otimes (\mathbf{I}+\mathbf{W}_g^2)).
\end{aligned}
$$

By combining all the above, we obtain a single ODE for $(\boldsymbol{\xi}, \boldsymbol{\eta}, \boldsymbol{\zeta})$:

$$
\frac{\mathrm{d}}{\mathrm{d}t}
\begin{bmatrix} \boldsymbol{\xi} \\ \boldsymbol{\eta} \\ \boldsymbol{\zeta} \end{bmatrix}
= -
\underbrace{
\left[
\begin{array}{c|c|c}
3\rho\mathbf{I} + \mathbf{K}_{11} & \mathbf{K}_{12} & \mathbf{K}_{13} \\
\hline
\mathbf{K}_{21} & 3\rho\mathbf{I} + \mathbf{K}_{22} & \mathbf{K}_{23} \\
\hline
\mathbf{K}_{31} & \mathbf{K}_{32} & 3\rho\mathbf{I} + \mathbf{K}_{33}
\end{array}
\right]}_{:=3\rho\mathbf{I}+\mathbf{K}}
\underbrace{\begin{bmatrix} \boldsymbol{\xi} \\ \boldsymbol{\eta} \\ \boldsymbol{\zeta} \end{bmatrix}}_{:=\boldsymbol{\Xi}},
$$

or alternatively, $\dot{\boldsymbol{\Xi}} = -(3\rho\mathbf{I} + \mathbf{K})\boldsymbol{\Xi}$. Note that $\mathbf{K}(t)$ is time-dependent. Finally, we can obtain the desired result by invoking Tian et al. (2021, Lemma 2).

**Lemma 1** (Tian et al. (2021, Lemma 2)). *Let* $\mathbf{H}(t)$ *be time-varying positive semidefinite matrices whose minimal eigenvalues are bounded away from zero:*

$$\inf_{t \geq 0} \lambda_{\min}(\mathbf{H}(t)) \geq \lambda_0 > 0.$$

*Then, the following dynamics*

$$\frac{\mathrm{d}\mathbf{w}(t)}{\mathrm{d}t} = -\mathbf{H}(t)\mathbf{w}(t)$$

*satisfies* $\|\mathbf{w}(t)\|_2 \leq \exp(-\lambda_0 t)\|\mathbf{w}(0)\|_2$, *which means that* $\mathbf{w}(t) \to \mathbf{0}$.

When minimal eigenvalues of $3\rho\mathbf{I} + \mathbf{K}(t)$ are always bounded away from zero, we immediately see $\mathbf{\Xi}(t) \to \mathbf{0}$, namely, $(\mathbf{C}_1(t), \mathbf{C}_2(t), \mathbf{C}_3(t)) \to (\mathbf{O}, \mathbf{O}, \mathbf{O})$ as $t \to \infty$. The strict positive-definiteness of $3\rho\mathbf{I} + \mathbf{K}(t)$ would not be necessarily satisfied; however, larger weight decay $\rho > 0$ induces it more easily. The convergence of the commutators is faster with larger $\rho > 0$ as well.

## B.3 Decoupling into Eigenvalue Dynamics

We have obtained the following matrix dynamics:

$$\dot{\mathbf{W}}_f = -\mathbf{W}_h^\top \{(1 + \sigma^2)(\mathbf{I} + \mathbf{W}_g^\top \mathbf{W}_g)\mathbf{W}_h - (\mathbf{I} + \mathbf{W}_g^\top)\}\mathbf{W}_f - \rho\mathbf{W}_f,$$
$$\dot{\mathbf{W}}_g = -\{(1 + \sigma^2)\mathbf{W}_h - \mathbf{I}\}\mathbf{W}_f \mathbf{W}_f^\top \mathbf{W}_h^\top - \rho\mathbf{W}_g,$$
$$\dot{\mathbf{W}}_h = -\{(1 + \sigma^2)(\mathbf{I} + \mathbf{W}_g^\top \mathbf{W}_g)\mathbf{W}_h - (\mathbf{I} + \mathbf{W}_g^\top)\}\mathbf{W}_f \mathbf{W}_f^\top - \rho\mathbf{W}_h.$$

Our aim is to decouple the matrix dynamics into their eigenvalue counterparts. Beforehand, let us execute the change-of-variable $\mathbf{\Phi} = \mathbf{W}_f \mathbf{W}_f^\top$:

$$\dot{\mathbf{\Phi}} = \dot{\mathbf{W}}_f \mathbf{W}_f^\top + \mathbf{W}_f \dot{\mathbf{W}}_f^\top$$
$$= -\mathbf{W}_h^\top \left\{(1 + \sigma^2)(\mathbf{I} + \mathbf{W}_g^\top \mathbf{W}_g)\mathbf{W}_h - (\mathbf{I} + \mathbf{W}_g^\top)\right\} \mathbf{\Phi}$$
$$- \mathbf{\Phi} \left\{(1 + \sigma^2)\mathbf{W}_h^\top(\mathbf{I} + \mathbf{W}_g^\top \mathbf{W}_g) - (\mathbf{I} + \mathbf{W}_g)\right\} \mathbf{W}_h - 2\rho\mathbf{\Phi}.$$

By the symmetry assumption (A1), $(\mathbf{\Phi}, \mathbf{W}_g, \mathbf{W}_h)$-dynamics can be simplified as follows:

$$\dot{\mathbf{\Phi}} = -(1 + \sigma^2)\{\mathbf{W}_h(\mathbf{I} + \mathbf{W}_g^2)\mathbf{W}_h, \mathbf{\Phi}\} + \{\mathbf{W}_h, \mathbf{\Phi}\} + (\mathbf{W}_h \mathbf{W}_g \mathbf{\Phi} + \mathbf{\Phi}\mathbf{W}_g \mathbf{W}_h) - 2\rho\mathbf{\Phi}, \quad (5)$$
$$\dot{\mathbf{W}}_g = -\left\{(1 + \sigma^2)\mathbf{W}_h - \mathbf{I}\right\}\mathbf{\Phi}\mathbf{W}_h - \rho\mathbf{W}_g,$$
$$\dot{\mathbf{W}}_h = -\left\{(1 + \sigma^2)(\mathbf{I} + \mathbf{W}_g^2)\mathbf{W}_h + (\mathbf{I} + \mathbf{W}_g)\right\}\mathbf{\Phi} - \rho\mathbf{W}_h,$$

where $\{\mathbf{A}, \mathbf{B}\} := \mathbf{AB} + \mathbf{BA}$ is the anticommutator for two symmetric matrices $\mathbf{A}$ and $\mathbf{B}$ with the same dimension.

Next, we decouple them into the corresponding eigenvalues. The parameter matrices are simultaneously diagonalized by $\mathbf{\Phi} = \mathbf{U}\mathbf{\Lambda}_\Phi \mathbf{U}^\top$, $\mathbf{W}_g = \mathbf{U}\mathbf{\Lambda}_g \mathbf{U}^\top$, and $\mathbf{W}_h = \mathbf{U}\mathbf{\Lambda}_h \mathbf{U}^\top$, with the aid of the symmetry assumption (A1) and common eigenspace assumption (A2). Here, we can easily show that the eigenspace is time-independent, namely, $\dot{\mathbf{U}} = \mathbf{O}$, using the same argument of Tian et al. (2021, Appendix B.1). By multiplying $\mathbf{U}^\top$ and $\mathbf{U}$ from left and right, respectively, $\mathbf{\Phi}$-dynamics can be written as follows:

$$\dot{\mathbf{\Lambda}}_\Phi = -2(1 + \sigma^2)(\mathbf{I} + \mathbf{\Lambda}_g^2)\mathbf{\Lambda}_h^2 \mathbf{\Lambda}_\Phi + 2\mathbf{\Lambda}_h \mathbf{\Lambda}_\Phi + 2\mathbf{\Lambda}_h \mathbf{\Lambda}_g \mathbf{\Lambda}_\Phi - 2\rho\mathbf{\Lambda}_\Phi,$$

where all matrices are diagonal, and thus, we can write down the dynamics in terms of $j$-th diagonal element (but the index $j$ is omitted for simplicity):

$$\dot{\phi} = -2(1 + \sigma^2)(1 + \gamma^2)\psi^2 \phi + 2\psi\phi + 2\psi\phi\gamma - 2\rho\phi.$$

We can decouple $\mathbf{W}_g$- and $\mathbf{W}_h$-dynamics similarly:

$$\dot{\gamma} = -(1 + \sigma^2)(\psi - 1)\phi\psi - \rho\gamma,$$
$$\dot{\psi} = -\{(1 + \sigma^2)(1 + \gamma^2)\psi + (1 + \gamma)\}\phi - \rho\psi.$$

Note that $\psi$ and $\gamma$ correspond to (one of the) eigenvalues of the linear networks $h$ and $g$, respectively. Intuitively, we can regard $\psi$ and $\gamma$ as "scalarization" of the predictor networks.

To sum it up, we decouple the dynamics of $(\mathbf{\Phi}, \mathbf{W}_h, \mathbf{W}_g)$ into the following dynamics of $(\phi, \psi, \gamma)$:

$$(\mathbf{\Phi}\text{-dynamics}) \qquad \dot{\phi} = -2\psi\phi\{(1+\sigma^2)(1+\gamma^2)\psi - (1+\gamma)\} - 2\rho\phi, \qquad (6)$$

$$(\mathbf{W}_h\text{-dynamics}) \qquad \dot{\psi} = -\phi\{(1+\sigma^2)(1+\gamma^2)\psi - (1+\gamma)\} - \rho\psi, \qquad (7)$$

$$(\mathbf{W}_g\text{-dynamics}) \qquad \dot{\gamma} = -\psi\phi\{(1+\sigma^2)\psi - 1\} - \rho\gamma, \qquad (8)$$

### B.4 ADIABATIC ELIMINATION

The eigenvalue dynamics obtained in Appendix B.3 is jointly with respect to $(\phi, \psi, \gamma)$. Here, we eliminate $\phi$ by confirming that $\phi$ and $\psi$ are asymptotically bound on an *invariant parabola*.

By combining (6) and (7), we have $2\psi\dot{\psi} - \dot{\phi} = -2\rho(\psi^2 - \phi)$. This can be integrated, and we obtain the following solution:

$$\psi(t)^2 - \phi(t) = C\exp(-2\rho t) \overset{t\to\infty}{\to} 0, \qquad (9)$$

where $C$ is a constant of integration. Thus, $(\phi(t), \psi(t))$ converges to this invariant parabola (9) exponentially quickly, which we suppose is much faster than the dynamics stabilization. On this invariant parabola $\phi = \psi^2$, the eigenvalue dynamics can be further simplified as follows by eliminating $\phi$:

$$\begin{cases} \dot{\psi} &= \{(1+\gamma) - (1+\sigma^2)(1+\gamma^2)\psi\}\psi^2 - \rho\psi, \\ \dot{\gamma} &= \{1 - (1+\sigma^2)\psi\}\psi^3 - \rho\gamma. \end{cases}$$

Note that the convergence to the invariant parabola is faster when weight decay $\rho$ is more intense.

## C PSEUDOCODE FOR PHINET AND X-PHINET

The pseudo codes for PhiNet and X-PhiNet are shown in Listing.1 and Listing.2.

```
1  # f: backbone + projection mlp
2  # h: prediction mlp
3  # g: prediction mlp
4
5  for x in loader: # load a minibatch x with n samples
6      x1, x2 = aug(x), aug(x) # random augmentation
7      z0, z1, z2 = f(x), f(x1), f(x2) # projections, n-by-d
8      p1, p2 = h(z1), h(z2) # predictions, n-by-d
9      y1, y2 = g(p1), g(p2) # predictions, n-by-d
10     z0 = z0.detach()
11     Lcos = D(p1, z2)/2 + D(p2, z1)/2 # loss
12     Lcor = mse_loss(y1, z0)/2 + mse_loss(y2,z0)/2
13     L = Lcos + Lcor
14     L.backward() # back-propagate
15     update(f, h) # SGD update
16
17 def D(p, z): # negative cosine similarity
18     z = z.detach() # stop gradient
19     p = normalize(p, dim=1) # l2-normalize
20     z = normalize(z, dim=1) # l2-normalize
21     return -(p*z).sum(dim=1).mean()
```

Listing 1: PhiNet Pseudocode (PyTorch-like)

```
1  # f: backbone + projection mlp
2  # h: prediction mlp
3  # g: prediction mlp
4
5  for x in loader: # load a minibatch x with n samples
6      x1, x2 = aug(x), aug(x) # random augmentation
7      z0, z1, z2 = f_long(x), f(x1), f(x2) # projections, n-by-d
8      p1, p2 = h(z1), h(z2) # predictions, n-by-d
9      y1, y2 = g(p1), g(p2) # predictions, n-by-d
10     z0 = z0.detach()
```

Table 3: **X-PhiNet performs robustly for different weight decays on CIFAR-5m.**

| | Accuracy by Linear Probing (w.r.t. weight decay) | | | |
| --- | --- | --- | --- | --- |
| | 0.0001 | 5e-05 | 2e-05 | 1e-05 |
| BYOL | $67.88_{0.58}$ | $75.71_{0.34}$ | $81.05_{0.04}$ | $80.70_{0.84}$ |
| SimSiam | $77.69_{0.67}$ | $75.02_{5.92}$ | $76.87_{3.13}$ | $77.71_{1.97}$ |
| PhiNet | $76.43_{2.12}$ | $77.57_{2.01}$ | $77.64_{1.44}$ | $77.74_{0.79}$ |
| RM-SimSiam | $74.24_{0.56}$ | $77.52_{0.88}$ | $82.09_{0.22}$ | $79.38_{0.38}$ |
| X-PhiNet with Aug (mse) | $65.96_{16.24}$ | $83.21_{0.15}$ | $86.45_{0.25}$ | $85.17_{0.54}$ |
| X-PhiNet with Aug (cos) | $84.31_{0.29}$ | $86.40_{0.31}$ | $86.96_{0.17}$ | $84.85_{0.99}$ |
| X-PhiNet (mse) | $69.02_{14.25}$ | $84.24_{0.37}$ | $\mathbf{87.30}_{0.13}$ | $85.11_{0.17}$ |
| X-PhiNet (cos) | $85.80_{0.34}$ | $87.29_{0.22}$ | $\mathbf{87.46}_{0.19}$ | $85.03_{0.19}$ |
| X-PhiNet (cos, $g = I$) | $84.72_{0.16}$ | $86.41_{0.08}$ | $86.71_{0.27}$ | $83.83_{0.36}$ |

Table 4: **X-PhiNet performs robustly well for one epoch training.**

| | Accuracy by Linear Probing (w.r.t. weight decay) | | | |
| --- | --- | --- | --- | --- |
| | 0.0001 | 5e-05 | 2e-05 | 1e-05 |
| BYOL | $63.86_{0.77}$ | $59.80_{0.50}$ | $58.04_{0.52}$ | $57.65_{0.38}$ |
| SimSiam | $68.50_{0.19}$ | $69.65_{0.29}$ | $69.41_{1.06}$ | $69.60_{0.97}$ |
| PhiNet | $66.27_{1.60}$ | $64.26_{1.82}$ | $64.34_{1.13}$ | $62.68_{1.60}$ |
| RM-SimSiam | $62.90_{1.11}$ | $63.30_{1.86}$ | $63.45_{0.92}$ | $63.05_{1.76}$ |
| X-PhiNet (mse) | $74.25_{0.80}$ | $72.65_{0.85}$ | $71.20_{0.48}$ | $71.95_{0.65}$ |
| X-PhiNet (cos) | $74.76_{0.52}$ | $72.89_{0.66}$ | $72.10_{0.15}$ | $71.93_{0.51}$ |

```
11    Lcos = D(p1, z2)/2 + D(p2, z1)/2 # loss
12    Lcor = mse_loss(y1, z0)/2 + mse_loss(y2,z0)/2
13    L = Lcos + Lcor
14    L.backward() # back-propagate
15    update(f, h) # SGD update
16    f_long = beta * f_long + (1-beta) * f # EMA for projection
17
18 def D(p, z): # negative cosine similarity
19    z = z.detach() # stop gradient
20    p = normalize(p, dim=1) # l2-normalize
21    z = normalize(z, dim=1) # l2-normalize
22    return -(p*z).sum(dim=1).mean()
```

Listing 2: X-PhiNet Pseudocode (PyTorch-like)

# D ADDITIONAL EXPERIMENTS ON ONLINE AND CONTINUAL LEARNING

## D.1 CIFAR-5M (ONLINE LEARNING)

Table 3 shows the accuracy of linear probing for different weight decay values in CIFAR-5m. X-PhiNet consistently demonstrates high performance. In Table 4, we trained for only one epoch on CIFAR-5m. Also in this case, X-PhiNet performs better than SimSiam. Additionally, X-PhiNet with $g = I$ achieves lower accuracy compared to the standard X-PhiNet (cos). This represents a major difference from the results shown in Figure 6, where both achieved similar accuracy with batch size = 128.

In Table 3, "X-PhiNet with Aug (mse)" and "X-PhiNet with Aug (cos)" represent cases where data augmentation is also applied to the input $x$ of $f_{\mathrm{long}}$. In this scenario, all inputs to the model are augmented. While X-PhiNet with Aug outperforms other baselines such as SimSiam and RM-SimSiam, its performance is still inferior to our standard X-PhiNet. This discrepancy might be attributed to an imbalance in regularization strength, although the exact reason remains unclear and is left for future work.

## D.2 CONTINUAL LEARNING

**Epochs per task.** In Table 2, we trained on each task for one epoch. However, in Madaan et al. (2022), 200 epochs were trained for each task on Split CIFAR10, and the number of iterations differs from this case. The effect of early stopping may be apparent when the number of iterations is different. Thus, we trained on each task for two epochs to match the number of iterations. The result is shown in Table 5. The performance of X-PhiNet is still high even when the number of epochs per task is set to 2 epochs.

Table 5: **X-PhiNet shows higher accuracy when the number of epochs per task is increased.** We trained X-PhiNet on Split CIFAR-5m. Unlike Table 2, this table presents results obtained from training 2 epochs for each task.

|  |  | BYOL | SimSiam | Barlow Twins | PhiNet | RM-SimSiam | X-PhiNet (MSE) | X-PhiNet (Cos) |
|---|---|---|---|---|---|---|---|---|
| Split C-5m | Acc | $91.36_{0.25}$ | $92.25_{0.10}$ | $90.73_{0.28}$ | $92.22_{0.09}$ | $90.11_{0.34}$ | $92.33_{0.15}$ | $92.83_{0.06}$ |
| (2epoch) | Fg | $4.10_{0.25}$ | $3.88_{0.33}$ | $5.25_{0.67}$ | $4.01_{0.26}$ | $6.12_{0.50}$ | $3.79_{0.47}$ | $3.71_{0.14}$ |

**Replay with Mixup.** Replay is one of the most promising methods for improving the performance of continual learning though it requires additional memory costs (Hsu et al., 2018; Van de Ven et al., 2020; Madaan et al., 2022; Lin et al., 2022). We thus examined the performance of our method in combination with the mixup-based replay method proposed in (Madaan et al., 2022). Table 6 shows that when only one epoch is trained for each task, X-PhiNet shows considerably higher accuracy than the other methods. On the other hand, when we train two epochs for each task, the accuracy of other methods such as BYOL, BarlowTwins and RM-SimSiam also increases, showing an accuracy comparable to that of X-PhiNet.

Table 6: **X-PhiNet performs higher or comparable results for Split-CIFAR5m even with Mixup.** We trained X-PhiNet on Split CIFAR-5m with replay methods.

|  |  | BYOL | SimSiam | Barlow Twins | PhiNet | RM-SimSiam | X-PhiNet (MSE) | X-PhiNet (Cos) |
|---|---|---|---|---|---|---|---|---|
| Split C-5m | Acc | $90.18_{0.65}$ | $91.51_{0.42}$ | $90.74_{0.63}$ | $91.66_{0.21}$ | $91.92_{0.17}$ | $91.78_{0.29}$ | $\mathbf{92.43_{0.14}}$ |
| (1epoch) | Fg | $0.36_{3.07}$ | $-1.70_{0.09}$ | $-0.97_{3.05}$ | $-2.21_{0.54}$ | $-1.14_{0.27}$ | $-0.73_{0.07}$ | $-0.69_{0.65}$ |
| Split C-5m | Acc | $\mathbf{92.36_{0.03}}$ | $91.77_{0.01}$ | $\mathbf{92.36_{0.70}}$ | $91.00_{1.78}$ | $\mathbf{92.48_{0.12}}$ | $92.12_{0.28}$ | $92.26_{0.35}$ |
| (2epoch) | Fg | $0.64_{0.01}$ | $2.24_{0.01}$ | $-0.07_{0.64}$ | $1.97_{1.41}$ | $1.05_{0.31}$ | $2.02_{0.71}$ | $1.82_{0.71}$ |

**Split CIFAR10 and Split CIFAR100.** Up to this point, we have experimented with continual learning using the CIFAR-5m-based dataset. Now, we test on the standard benchmarks, Split CIFAR10 and Split CIFAR100. Table 7 shows that in both Split CIFAR10 and Split CIFAR100, X-PhiNet outperforms SimSiam. However, PhiNet sometimes shows higher accuracy than X-PhiNet. Note that PhiNet is a special case of X-PhiNet, and we have set the momentum of X-PhiNet to 0.99 in this study. If we carefully select the momentum value, X-PhiNet's performance might improve, surpassing PhiNet. When using mixup for replay, X-PhiNet shows significantly higher accuracy compared to other methods.

**Effect of exponential moving average** X-PhiNet has an additional hyperparameter, the exponential moving average. We set $\beta = 0.99$ in all the experiments in this paper. As shown in Table 8, in tasks such as continual learning, where it is important to apply a strong exponential moving average, accuracy increases as $\beta$ increases and then decreases again from a certain point.

## E ADDITIONAL EXPERIMENTS ON THE ROBUSTNESS OF PHINET

### E.1 ADDITIONAL ABLATION STUDY WITH CIFAR10

**Usage of original input:** We first investigate what is the best way to input the original signal to the model. To this end, we first replace one of the augmented signals in SimSiam as an original

Table 7: **X-PhiNet produces good results when memorization is important.** We trained X-PhiNet on Split CIFAR10. Acc is the average of the final Acc (higher is better), and Fg is Forgetting (smaller is better).

|  |  | SimSiam | RM-SimSiam | PhiNet (MSE) | X-PhiNet ($g = I$, MSE) | X-PhiNet (MSE) | X-PhiNet (Cos) |
|---|---|---|---|---|---|---|---|
| Split C10 (FineTune) | Acc | $91.05_{0.29}$ | $89.35_{0.08}$ | $\mathbf{91.25_{0.09}}$ | $90.65_{0.43}$ | $90.90_{0.50}$ | $\mathbf{90.97_{0.49}}$ |
|  | Fg | $5.31_{0.65}$ | $3.70_{0.16}$ | $4.86_{0.50}$ | $1.02_{0.31}$ | $5.72_{0.82}$ | $3.95_{0.37}$ |
| Split C100 (FineTune) | Acc | $77.93_{0.64}$ | $78.19_{0.41}$ | $\mathbf{78.50_{0.25}}$ | $78.31_{0.16}$ | $77.50_{0.04}$ | $77.44_{0.28}$ |
|  | Fg | $7.06_{1.00}$ | $-0.57_{0.98}$ | $6.51_{0.31}$ | $5.49_{1.27}$ | $8.46_{0.21}$ | $4.46_{0.34}$ |
| Split C10 (Mixup) | Acc | $90.68_{0.89}$ | $91.14_{0.84}$ | $89.89_{0.69}$ | $90.69_{0.21}$ | $90.49_{0.32}$ | $\mathbf{91.56_{0.12}}$ |
|  | Fg | $0.85_{0.16}$ | $1.08_{0.60}$ | $1.04_{0.22}$ | $-2.11_{1.61}$ | $1.80_{0.09}$ | $1.36_{0.15}$ |
| Split C100 (Mixup) | Acc | $81.77_{0.14}$ | $82.47_{0.70}$ | $80.76_{0.18}$ | $82.16_{0.76}$ | $83.32_{0.04}$ | $\mathbf{83.88_{0.26}}$ |
|  | Fg | $1.23_{0.81}$ | $-1.35_{0.05}$ | $1.18_{1.27}$ | $-1.23_{2.11}$ | $1.28_{0.34}$ | $-0.07_{0.30}$ |

Table 8: **X-PhiNet produces good results when memorization is important.** We trained X-PhiNet on Split CIFAR100 with mixup with different exponential moving average value.

|  |  | EMA $\beta$ | | | | | |
|---|---|---|---|---|---|---|---|
|  |  | 0.999 | 0.997 | 0.99 | 0.97 | 0.9 | 0.7 |
| Split C100 (Mixup) | Acc | $83.19_{0.22}$ | $82.62_{0.06}$ | $83.88_{0.26}$ | $82.72_{0.98}$ | $81.76_{0.64}$ | $81.73_{0.06}$ |
|  | Fg | $-0.22_{0.34}$ | $0.33_{0.06}$ | $-0.07_{0.30}$ | $0.73_{0.64}$ | $2.15_{0.11}$ | $1.82_{0.69}$ |

input. Then, we found that comparing the augmented images and original input significantly degrades the model performance. This indicates that the original SimSiam performs pretty well even if we do not use the original inputs, and naively adding additional input hurts the model performance significantly. In contrast, the PhiNet with MSE loss and StopGradient compares favorably with the original SimSiam model.

**StopGradient-2:** We analysed the impact of the StopGradient-2 technique, as shown in the table. The StopGradient operator effectively prevents mode collapse. Interestingly, while the StopGradient operator is not essential for avoiding mode collapse, models without it perform worse compared to those with it. Thus, the StopGradient operator contributes to improved stability when using the MSE loss. On the other hand, mode collapse still occurs with the negative cosine loss function.

### E.2 COMPARISON OF FAST LEARNER WITH SLOW LEARNER

We can observe that EMA plays as a slow learner through experiments. In Figure 9, we conducted continual learning experiments, where linear probing of the EMA encoder (dashed lines) performs consistently worse than the encoder without EMA (solid lines). This indicates that EMA does not quickly adapt to the most recent samples and learns more stable features as a slow learner. In this sense, we believe it is natural to think of slow learning as serving as a regularization for past samples, similar to other continual learning techniques such as elastic weight consolidation.

### E.3 ADDITIONAL ABLATION STUDY WITH IMAGENET

Table 10 shows the ablation of additional predictors in ImageNet. In this case, we used a higher weight decay of 1e-3. $g = h$ has a lower accuracy than other methods, which is consistent with the results in CIFAR10.

### E.4 FOR DIFFERENT DATASETS AND EVALUATION METRICS WITH DIFFERENT BATCH SIZES

Table 11 and Table 12 present the CIFAR10 experiment results with varying batch sizes and weight decay. PhiNet shows equal or better performance than SimSiam across different batch sizes. The evaluation trends from KNN classification and linear probing are also consistent. It is also a consistent result that training on CIFAR10 performs poorly when cosine loss is used as the cortex loss function. Table 13 shows the results for STL10, where PhiNet performs comparably to SimSiam. However, as

Table 9: Ablation study for PhiNet using CIFAR10 data. We use SGD with momentum as an optimiser and set the base learning rate as 0.03 and run 800 epochs. We evaluated the performance using KNN classification with $K = 200$. See Table 19 for further details.

| Method | Sim-2 | SG-2 | Pred-2 | Acc (w.r.t. weight decay) | | |
|---|---|---|---|---|---|---|
| | | | | 0.0 | 0.0005 | 0.001 |
| SimSiam | – | – | – | $74.12_{0.39}$ | $90.39_{0.10}$ | $90.98_{0.02}$ |
| SimSiam (Orig-In) | – | – | – | $72.82_{0.18}$ | $76.67_{1.13}$ | $69.03_{11.37}$ |
| PhiNet | MSE | ✓ | $g$ | $77.77_{1.13}$ | $90.77_{0.22}$ | $91.38_{0.19}$ |
| | MSE | ✓ | $g = \mathbf{I}$ | $77.63_{0.11}$ | $91.01_{0.12}$ | $91.50_{0.07}$ |
| | MSE | | $g$ | $62.80_{0.47}$ | $91.40_{0.23}$ | $89.01_{0.55}$ |
| | MSE | ✓ | $h$ | $74.87_{0.58}$ | $91.23_{0.12}$ | $91.18_{0.34}$ |
| | Cos | ✓ | $g$ | $80.06_{0.47}$ | $87.73_{0.26}$ | $88.27_{0.24}$ |
| | Cos | ✓ | $g = \mathbf{I}$ | $75.34_{3.27}$ | $87.38_{0.17}$ | $87.90_{0.10}$ |
| | Cos | | $g$ | $27.57_{4.41}$ | $9.98_{0.00}$ | $9.98_{0.00}$ |
| | Cos | ✓ | $h$ | $75.99_{0.28}$ | $85.97_{0.23}$ | $85.04_{0.11}$ |

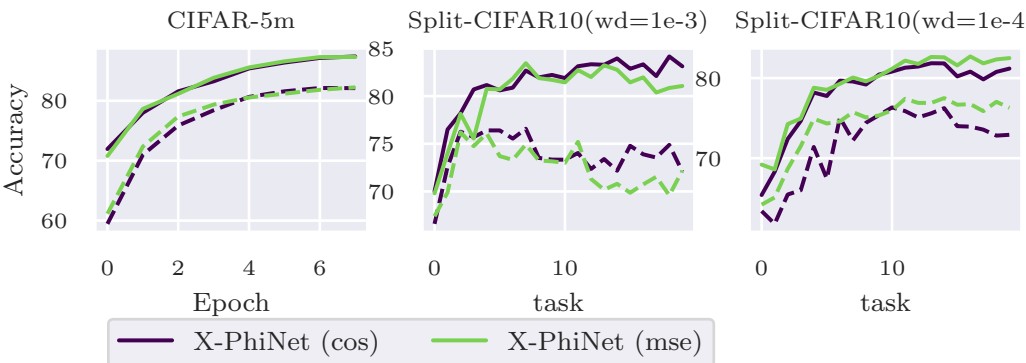

Figure 9: **The accuracy for slow weight is lower than the accuracy for fast weight.** The solid line represents the accuracy of the fast weights (the encoder without momentum), while the dashed line represents the accuracy of the slow weights (the encoder with momentum). Note that the accuracy for split-CIFAR10 represents the average accuracy.

Table 10: $g = h$ **has a low accuracy on ImageNet.** We trained the models for 100 epochs and then validated them on the test sets using linear probing on the head. Unlike Table 1, we train linear probing for 40 epochs to save computational costs.

| | Model | | | |
|---|---|---|---|---|
| | SimSiam | PhiNet (MSE) | PhiNet ($g = I$) | PhiNet ($g = h$) |
| Linear Probing Acc | 66.35 | 66.64 | 66.47 | 55.12 |

illustrated in Figure 7, PhiNet demonstrates better convergence in the early learning stages. In the early stages of training, when SimSiam is not stable, the cosine loss is smaller than that of the PhiNet, while as the training continues, the cosine loss increases again, in agreement with the PhiNet. This suggests that something close to mode collapse occurs in the early stages of SimSiam training, while PhiNet may suppress this collapse.

### E.5 PERFORMANCE ON TRANSFER LEARNING

We conducted experiments for transfer learning using object detection on VOC. In Table 14, following the original SimSiam paper, we conducted pre-training experiments with two different settings for learning rate and weight decay. This table demonstrates that our X-PhiNet produces comparable performance to MoCo across various tasks. Furthermore, as shown in Table 15, we can find that

Table 11: **PhiNet shows equal or better performance than SimSiam.** Sensitivity analysis for PhiNet using CIFAR10 data. We use SGD with momentum as an optimiser, set the base learning rate as 0.03, and run 800 epochs. We evaluated the performance using KNN classification with $K = 200$.

| weight decay | | Acc (w.r.t. batch size) | | | |
| --- | --- | --- | --- | --- | --- |
| | | 128 | 256 | 512 | 1024 |
| 0.0001 | SimSiam | $86.35_{2.28}$ | $88.05_{0.66}$ | $88.34_{0.28}$ | $85.96_{2.93}$ |
| | PhiNet | $\mathbf{88.90_{0.23}}$ | $\mathbf{88.92_{0.10}}$ | $\mathbf{88.93_{0.33}}$ | $\mathbf{88.91_{0.07}}$ |
| | X-PhiNet (MSE) | $\mathbf{88.67_{0.39}}$ | $\mathbf{88.67_{0.23}}$ | $\mathbf{88.71_{0.20}}$ | $88.44_{0.32}$ |
| | X-PhiNet (Cos) | $82.57_{5.67}$ | $79.31_{10.65}$ | $72.35_{19.83}$ | $84.79_{0.56}$ |
| 0.0005 | SimSiam | $\mathbf{90.15_{0.15}}$ | $\mathbf{90.36_{0.15}}$ | $90.39_{0.10}$ | $90.89_{0.08}$ |
| | PhiNet | $\mathbf{90.40_{0.16}}$ | $\mathbf{90.48_{0.34}}$ | $\mathbf{90.57_{0.15}}$ | $\mathbf{91.15_{0.04}}$ |
| | X-PhiNet (MSE) | $\mathbf{90.05_{0.29}}$ | $90.13_{0.02}$ | $90.39_{0.12}$ | $90.70_{0.10}$ |
| | X-PhiNet (Cos) | $86.35_{0.23}$ | $86.36_{0.16}$ | $86.42_{0.11}$ | $86.65_{0.65}$ |
| 0.001 | SimSiam | $\mathbf{91.23_{0.11}}$ | $91.30_{0.05}$ | $90.98_{0.02}$ | $76.68_{12.83}$ |
| | PhiNet | $\mathbf{91.23_{0.07}}$ | $\mathbf{91.44_{0.08}}$ | $\mathbf{91.50_{0.03}}$ | $73.97_{7.04}$ |
| | X-PhiNet (MSE) | $91.04_{0.13}$ | $91.09_{0.14}$ | $91.11_{0.13}$ | $\mathbf{90.08_{0.37}}$ |
| | X-PhiNet (Cos) | $86.54_{0.18}$ | $86.33_{1.04}$ | $86.79_{0.75}$ | $87.47_{1.16}$ |

Table 12: **Linear probing shows similar trends to knn classification.** Sensitivity analysis for PhiNet using CIFAR10 data. We use SGD with momentum as an optimiser, set the base learning rate as 0.03, and run 800 epochs. We evaluated the performance using linear probing on the head.

| weight decay | | Acc (w.r.t. batch size) | | | |
| --- | --- | --- | --- | --- | --- |
| | | 128 | 256 | 512 | 1024 |
| 0.0001 | SimSiam | $88.64_{1.73}$ | $89.44_{0.35}$ | $89.39_{0.49}$ | $88.01_{1.80}$ |
| | PhiNet | $90.27_{0.13}$ | $89.83_{0.35}$ | $89.79_{0.24}$ | $89.70_{0.21}$ |
| | X-PhiNet (MSE) | $88.67_{0.39}$ | $88.67_{0.23}$ | $88.71_{0.20}$ | $88.44_{0.32}$ |
| | X-PhiNet (Cos) | $83.29_{4.10}$ | $84.15_{0.46}$ | $72.35_{19.84}$ | $83.23_{2.52}$ |
| 0.0005 | SimSiam | $90.39_{0.07}$ | $90.68_{0.05}$ | $91.15_{0.12}$ | $91.65_{0.06}$ |
| | PhiNet | $90.68_{0.10}$ | $90.87_{0.34}$ | $91.11_{0.08}$ | $91.93_{0.13}$ |
| | X-PhiNet (MSE) | $90.05_{0.29}$ | $90.13_{0.02}$ | $90.39_{0.12}$ | $90.70_{0.10}$ |
| | X-PhiNet (Cos) | $86.52_{0.12}$ | $86.47_{0.11}$ | $86.45_{0.16}$ | $87.00_{0.18}$ |
| 0.001 | SimSiam | $92.09_{0.22}$ | $92.36_{0.11}$ | $92.46_{0.29}$ | $77.71_{12.73}$ |
| | PhiNet | $92.18_{0.06}$ | $92.44_{0.21}$ | $92.63_{0.08}$ | $75.18_{6.69}$ |
| | X-PhiNet (MSE) | $91.04_{0.13}$ | $91.09_{0.14}$ | $91.11_{0.13}$ | $90.08_{0.37}$ |
| | X-PhiNet (Cos) | $87.04_{0.33}$ | $87.08_{0.20}$ | $87.24_{0.19}$ | $88.15_{0.09}$ |

Table 13: **SimSham and PhiNet show comparable performance.** Sensitivity analysis for PhiNet using STL10 data. We use SGD with momentum as an optimiser, set the base learning rate as 0.03, and run 800 epochs. We evaluated the performance using linear probing on the head.

| weight decay | | Acc (w.r.t. batch size) | | | |
| --- | --- | --- | --- | --- | --- |
| | | 128 | 256 | 512 | 1024 |
| 0.0001 | SimSiam | $85.97_{2.46}$ | $87.26_{0.26}$ | $87.53_{0.20}$ | $87.17_{0.05}$ |
| | PhiNet | $84.28_{0.41}$ | $87.32_{0.16}$ | $87.22_{0.12}$ | $87.01_{0.28}$ |
| 0.0005 | SimSiam | $88.89_{0.34}$ | $89.23_{0.11}$ | $89.39_{0.12}$ | $88.57_{0.40}$ |
| | PhiNet | $89.33_{0.13}$ | $89.26_{0.02}$ | $89.36_{0.27}$ | $88.62_{0.36}$ |
| 0.001 | SimSiam | $89.54_{0.05}$ | $89.61_{0.11}$ | $89.37_{0.03}$ | $nan_{nan}$ |
| | PhiNet | $89.52_{0.06}$ | $89.71_{0.07}$ | $89.28_{0.23}$ | $10.00_{0.01}$ |

| Pretrained | VOC 07 detection | | | VOC 07+12 detection | | |
|---|---|---|---|---|---|---|
| | $AP_{50}$ | AP | $AP_{75}$ | $AP_{50}$ | AP | $AP_{75}$ |
| MoCo v2 | 73.2 | **46.6** | 50.2 | 82.3 | 57.1 | 63.2 |
| SimSiam (lr=0.05, wd=1e-4) | 71.7 | 45.5 | 49.4 | 80.6 | 55.1 | 61.0 |
| SimSiam (lr=0.5, wd=1e-5) | 73.6 | **46.6** | 49.8 | **82.7** | **57.3** | 64.6 |
| PhiNet (lr=0.05, wd=1e-4) | 72.7 | 46.35 | **50.4** | 81.9 | 56.8 | 62.81 |
| PhiNet (lr=0.5, wd=1e-5) | 74.4 | 46.2 | 49.8 | 82.6 | 56.4 | 62.6 |
| X-PhiNet (lr=0.05, wd=1e-4) | 72.9 | 46.2 | 49.9 | 82.3 | 56.9 | **63.9** |
| X-PhiNet (lr=0.5, wd=1e-5) | **74.9** | 45.9 | 50.1 | **82.7** | 55.7 | 62.4 |

Table 14: **In transfer learning for object detection, X-PhiNet is comparable to MoCo (He et al., 2020) and SimSiam.** PhiNet is pre-trained by two training recipes similar to those in the SimSiam paper.

| wd | 2e-3 | 1e-3 | 5e-4 | 2.5e-4 | 1.25e-4 | 6.25e-5 | 3.125e-5 | 1.5625e-5 |
|---|---|---|---|---|---|---|---|---|
| SimSiam | 10.00 | 63.53 | 90.80 | 90.05 | 89.06 | 79.70 | 78.14 | 76.35 |
| MoCo | 87.47 | 88.11 | 87.76 | 87.01 | 85.79 | 83.45 | 81.34 | 80.44 |
| PhiNet | 10.00 | 78.33 | 91.19 | 90.35 | 89.34 | 86.25 | 83.21 | 81.60 |

Table 15: **PhiNet is robust to weight decay in transfer learning.** Performance comparison of SimSiam, MoCo, and PhiNet at different weight decay values.

PhiNet demonstrates a higher sensitivity to weight decay. Thus, it seems that our PhiNet can be extended to object detection without any modifications.

### E.6 ON THE AUGMENTATION FOR $x$

We use the unaugmented view for the Sim-2 loss to simulate a "time difference" between different views, which is partially supported by the temporal prediction hypothesis. This architecture does slightly increase the performance. See Table 16 and Figure 10, where "with aug" performs slightly worse than our proposed architecture while robustness to weight decay is still higher than SimSiam.

### E.7 COMPUTATIONAL COSTS

Table 17 shows the memory consumption when training on CIFAR10. There is little overhead for PhiNet and X-PhiNet over SimSiam, as the maximum memory consumption during training is not only related to weights, but also to gradients and activation state. In fact, the GPU consumption is highly dependent on batch size, indicating that the gradient and activation state, which are dependent on batch size, are dominant in this setting. Additionally, Table 18 includes a comparison of training times, demonstrating that PhiNet can be trained in time comparable to SimSiam.

| | Accuracy by Linear Probing (w.r.t. weight decay) | | | |
|---|---|---|---|---|
| | 0.0001 | 5e-05 | 2e-05 | 1e-05 |
| SimSiam | $77.69_{0.67}$ | $75.02_{5.92}$ | $76.87_{3.13}$ | $77.71_{1.97}$ |
| X-PhiNet with Aug (mse) | $65.96_{16.24}$ | $83.21_{0.15}$ | $86.45_{0.25}$ | $\mathbf{85.17_{0.54}}$ |
| X-PhiNet with Aug (cos) | $84.31_{0.29}$ | $86.40_{0.31}$ | $86.96_{0.17}$ | $84.85_{0.99}$ |
| X-PhiNet (mse) | $69.02_{14.25}$ | $84.24_{0.37}$ | $\mathbf{87.30_{0.13}}$ | $85.11_{0.17}$ |
| X-PhiNet (cos) | $\mathbf{85.80_{0.34}}$ | $\mathbf{87.29_{0.22}}$ | $\mathbf{87.46_{0.19}}$ | $85.03_{0.19}$ |

Table 16: **Even when $x$ is augmented, X-PhiNet performs better than SimSiam (CIFAR-5m).** We used the same setting as in Table 3 in the original paper. "with aug" performs data augmentation for x, which is not augmented in Table 3.

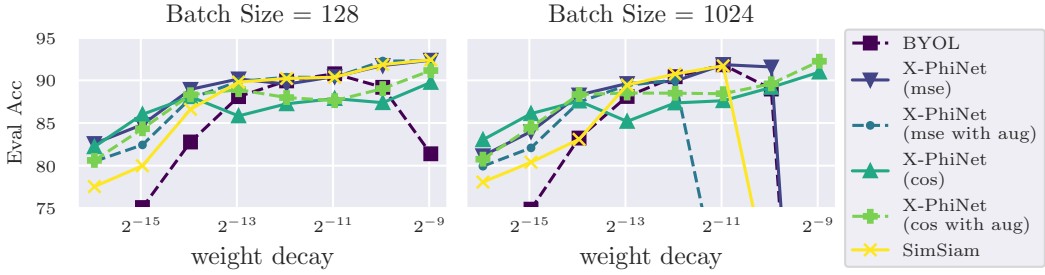

Figure 10: **Even when $x$ is augmented, PhiNet is more robust to weight decay than SimSiam (CIFAR10).** We used the same setting as in Figure 6 in the original paper. "with aug" performs data augmentation for x, which is not data augmented in Figure 6.

Table 17: **Comparison of GPU memory costs.** This is a comparison of memory consumption when training on CIFAR10. We report batch sizes of 128 and 1024.

| Batch Size | BYOL | SimSiam | Barlow Twins | PhiNet | RM-SimSiam | X-PhiNet (MSE) | X-PhiNet (Cos) |
|---|---|---|---|---|---|---|---|
| **BS=128** | 4.3 (GB) | 3.26 (GB) | 2.79 (GB) | 3.25 (GB) | 4.06 (GB) | 3.44 (GB) | 3.44 (GB) |
| **BS=1024** | 22.28 (GB) | 17.10 (GB) | 12.23 (GB) | 17.18 (GB) | 21.96 (GB) | 17.11 (GB) | 17.11 (GB) |

### E.8 STABLE RANK OF ADDITIONAL PREDICTOR LAYER

Figure 11 shows the rank for 2 linear layers in additional predictor blocks of PhiNet. We used stable rank in this figure and it is defined as $\mathrm{srank}(M) = \|M\|_F^2/\|M\|^2$, which is the lower rank of the standard rank and is more stable to the small eigenvalues of $M$. According to this figure, the rank of the additional layer remains large when the weight decay is small, suggesting that the additional layer may play a more important role in learning when the weight decay is small.

## F EXPERIMENTAL SETTINGS

### F.1 SETTINGS FOR TRAINING WITH CIFAR10, CIFAR100 AND STL10

Table 19 shows the model and experimental setup for Figure 6, Table 9, Table 11, Table 12 and Table 13. Note that in the graph of sensitivity with respect to weight decay, we explored a wider range of values. For linear probing evaluation, we trained the head layer by SGD for 100 epochs. For both CIFAR10 and STL10, we used 50,000 samples for training and 10,000 samples for testing. We have implemented it based on code that is already publicly available[1].

### F.2 SETTINGS FOR TRAINING ON IMAGENET

In our ImageNet (Russakovsky et al., 2015) experiments, we follow the formal implementation of SimSiam by Pytorch[2] (Chen and He, 2021). Table 20 shows the model and experimental setup for Table 1. For ImageNet, we used 1,281,167 samples for training and 100,000 samples for testing. We trained on the three seeds and obtained the mean and variance.

---

[1] https://github.com/PatrickHua/SimSiam
[2] https://github.com/facebookresearch/simsiam

Table 18: **Comparison of different models with varying batch sizes.**

| Batch Size | SimSiam | BYOL | Barlow-Twins | RM-SimSiam | PhiNet | X-PhiNet |
|---|---|---|---|---|---|---|
| **BS=128** | 6.89 (h) | 7.09 (h) | 21.38 (h) | 7.76 (h) | 6.89 (h) | 7.04 (h) |
| **BS=1024** | 6.74 (h) | 6.51 (h) | 7.68 (h) | 7.03 (h) | 6.44 (h) | 6.52 (h) |

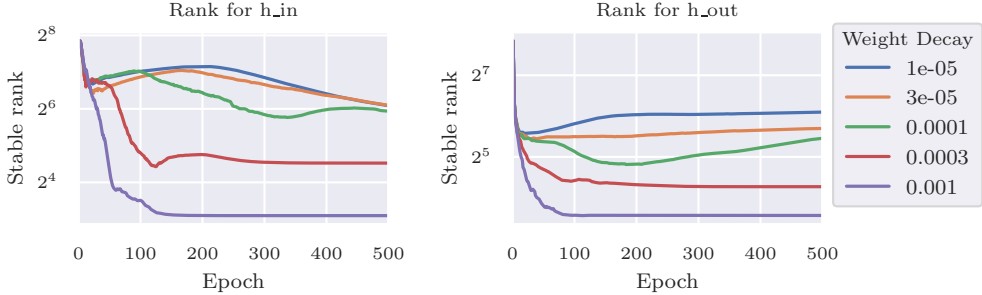

Figure 11: **The smaller the weight decay, the larger the rank of the additional predictor.** We trained PhiNet on CIFAR10 with SGD and evaluate the rank for layers in additional predictor blocks.

Table 19: **The experimental setups of Figure 6, Table 9, Table 11, Table 12 and Table 13.**

| | | |
|---|---|---|
| Learning | Optimiser | SGD |
| | Momentum | 0.9 |
| | Learning Rate | 0.03 |
| | Epochs | 800 |
| Encoder | Backbone | ResNet18_cifar_variant1 |
| | Projector output dimension | 2048 |
| Predictor $h$ | Latent dimension $m$ | 2048 |
| | Hidden dimension $h$ | 512 |
| | Activation function | ReLU |
| | Batch normalization | Yes |
| Predictor $g$ | Latent dimension $m$ | 2048 |
| | Hidden dimension $g$ | 512 |
| | Activation function (Hidden) | ReLU |
| | Activation function (Output) | Tanh |
| | Batch normalization | Yes |
| Computational resource | GPUs | V100 or A100 |

Table 20: **The experimental setups of Table 1.**

| | | |
|---|---|---|
| Learning | Optimiser | SGD |
| | Momentum | 0.9 |
| | Learning Rate | 0.05 |
| | Epochs | 100 |
| Encoder | Backbone | ResNet50 |
| | Projector output dimension | 2048 |
| Predictor $h$ | Latent dimension $m$ | 2048 |
| | Hidden dimension $h$ | 512 |
| | Activation function | ReLU |
| | Batch normalization | Yes |
| Predictor $g$ | Latent dimension $m$ | 2048 |
| | Hidden dimension $g$ | 512 |
| | Activation function (Hidden) | ReLU |
| | Activation function (Output) | Tanh |
| | Batch normalization | Yes |
| Computational resource | GPUs | 4×V100 |

### F.3 SETTINGS FOR TRAINING ON CIFAR-5M

CIFAR-5m (Nakkiran et al., 2021) is a dataset that is sometimes used as a vision dataset for online learning (Vyas et al., 2023; Sarnthein et al., 2023). We experimented with CIFAR-5m in a setting similar to online learning. Note that CIFAR-5m has 5m samples, but we chose to train CIFAR-5m for 8 epochs, as most of the SimSiam training on CIFAR10 involves training for 800 epochs. We experimented with three learning rates: {0.03, 0.01, 0.003}, and selected the one that yielded the best results. For Barlow Twins, the learning rate of 0.03 does not converge, so 0.003 is chosen instead. For all other methods, a learning rate of 0.03 is selected. The model architecture is the same as in CIFAR10.

### F.4 SETTINGS FOR TRAINING ON SPLIT CIFAR10, SPLIT CIFAR100 AND SPLIT-CIFAR-5M

As a benchmark for evaluating continual learning, we used split CIFAR10 and split CI-FAR100 (Krizhevsky, 2009). Additionally, we created split cifar-5m, which is inspired by split-CIFAR10 but uses CIFAR-5m dataset. In split CIFAR10 and split CIFAR5m, we split CIFAR10 and CIFAR-5m into 5 tasks, each of which contains 2 classes. In split CIFAR10 , we split CIFAR100 into 10 tasks, each of which contains 2 classes. The model architecture is the same as in CIFAR10. The implementation of continual learning is based on the official implementation of Madaan et al. (2022).

We evaluated the results using Average Accuracy and Average Forgetting. The average accuracy after the model has trained for $T$ tasks is defined as:

$$A_T = \frac{1}{T} \sum_{i=1}^{T} a_{T,i},\tag{10}$$

where $a_{t,i}$ is the validation accuracy on task $i$ after the model finished task $t$. The average forgetting is defined as the difference between the maximum accuracy and the final accuracy of each task. Therefore, average forgetting after the model has trained for $T$ tasks can be defined as:

$$F = \frac{1}{T-1} \sum_{i=1}^{T-1} \max_{t \in \{1,\dots,T-1\}} \left( a_{t,i} - a_{T,i} \right)\tag{11}$$

