# OpenReview forum: "PhiNets: Brain-inspired Non-contrastive Learning Based on Temporal Prediction Hypothesis"
_ICLR.cc/2025/Conference — ICLR 2025 Poster_

### Official Review · Reviewer_QH5V · 2024-10-26

**Soundness:** 3
**Presentation:** 3
**Contribution:** 4
**Rating:** 6
**Confidence:** 2

**Summary:**

This paper introduces PhiNet and X-PhiNet, where the input is processed in three parallel pathways with 0,1,2 predictors and with two pathways having the stop-gradient operation. The authors report favorable results over SimSiam and explore online/continual learning effectiveness in their models.

**Strengths:**

The idea is well motivated and the paper is thoroughly cited. The analysis is rigorous and convincing.

**Weaknesses:**

The majority of my weaknesses stem from my lack of understanding about certain claims, and I believe that this paper will be improved if the following points are made clearer:

- Can the authors explicitly explain, with specific reference to any important terms, why stop-gradient leads to any temporal lagging? I understand how stop-gradient works, but I fail to see how it is related to time in any way.

- I am not able to see any claims in section 5.1 in Figure 5, even though it is referenced in the text. What kind of improvement am I supposed to see and what effect am I looking for?

I will update my review once I gain a better understanding of these results.

**Questions:**

See weaknesses.

---

> ### Author Response · Authors · 2024-11-21
> **Response to Reviewer QH5V**
>
> We would like to thank the reviewer for the insightful questions and for correctly acknowledging the strengths of our work.
>
> ## Weaknesses
> > Can the authors explicitly explain, with specific reference to any important terms, why stop-gradient leads to any temporal lagging?
>
> Thank you for your question.
> First, our work introduces the novel claim that stop-gradient causes temporal lagging. Thus, there are no existing references that directly address this point.
> Please note that the temporal lagging mentioned in Section 3.1 refers specifically to the temporal lag in the weight update direction.
> To clarify, we are considering a loss between $f_t(x_i^{(1)})$ and $f_{t-1}(x_i^{(2)})$.
> In this loss function, $x_i^{(1)}$ is encoded with weights after $t$-step update while $x_i^{(2)}$ is encoded with weights after $t-1$-step update. This difference in encoding weights is what we refer to as temporal lag made by stop gradient.
> Note that, although it may be confusing, the temporal lagging between the image inputs $x_i^{(1)}$ and $x_i^{(2)}$ is caused by augmentation, not by temporal lagging in the weights.
>
> > I am not able to see any claims in section 5.1 in Figure 5, even though it is referenced in the text. What kind of improvement am I supposed to see and what effect am I looking for?
>
> Thank you for your question.
> In Figure 5, we aim to demonstrate two key points:
> 1. PhiNet (including X-PhiNet) achieves performance comparable to SimSiam.
> 2. Unlike BYOL or SimSiam, PhiNet (including X-PhiNet) exhibits minimal performance degradation when the weight decay is set lower than its optimal value.
>
> Regarding point (1), we observe that PhiNet (including X-PhiNet) achieves performance comparable to SimSiam. Furthermore, while RM-SimSiam employs EMA similarly to X-PhiNet, its performance on standard tasks tends to fall below that of PhiNet (and even below that of SimSiam), a limitation that PhiNet does not exhibit.
>
> For point (2), when the weight decay is smaller than $2^{-15}$, PhiNet (including X-PhiNet) achieves higher accuracy compared to both BYOL and SimSiam. This finding supports the theoretical analysis presented in Section 4.
> Furthermore, as discussed in "Bless of Additional CA1 Predictor," PhiNet with $g = h$ or $g = I$ performs worse than PhiNet with an additional predictor, particularly when the weight decay is small. The poor performance of $g = I$ is especially pronounced at a batch size of 1024.
> This suggests that PhiNet with an additional CA1 predictor appears more robust to weight decay, making it a preferable choice from the perspective of weight decay robustness.

---

> > ### Comment · Reviewer_QH5V · 2024-11-21
> > **Thank you**
> >
> > I thank the authors for their response. It is clear to me why there is temporal lagging now. I also understand Figure 5 more clearly now.
> >
> > My initial score already gave the authors the benefit of the doubt so I will keep my score.

---

### Official Review · Reviewer_1cHn · 2024-11-02

**Soundness:** 3
**Presentation:** 2
**Contribution:** 3
**Rating:** 8
**Confidence:** 4

**Summary:**

This papers takes another step in building non-contrastive self-supervised learning algorithms by taking inspiration from the architecture and supposed function of the hippocampal circuit. This PhiNets presented in this work are an extension of the SimSiam algorithm and the authors compare their method extensively to SimSiam, showing that it outperforms SimSiam in terms of robustness of low weight decay settings and continual learning. Their algorithm is competitive with many well-known contrastive and non-contrastive SSL methods (BYOL, Barlow Twins, etc.). Taken together, this is a nice paper showcasing the inspiration-from-neuroscience to usefully-deployed-ML-algorithm possibility.

**Strengths:**

Originality
- As far as I could tell, their conversion of a hippocampal circuit process to a ML algorithm is novel. It does go beyond the closest match, SimSiam, in both architecture and abilities.

Quality
- In addition to empirical results, the analytical treatment is super useful to make us understand *why* their algorithm might outperform SimSiam in terms of robustness to low weight decay (among other things).

Significance
- As I wrote in the summary, "this is a nice paper showcasing the inspiration-from-neuroscience to usefully-deployed-ML-algorithm possibility."

**Weaknesses:**

I have two core concerns:
1. The authors link their algorithm to the temporal prediction hypothesis. While I can see how in principle their setting (with image augmentations and the StopGradient) is similar to what temporal prediction would entail, results from this project cannot be used easily to validate their success as a success of temporal prediction SSL solely due to the reason that they do not use a temporal sequence. Yes, in a temporal sequence like a video subsequent frames could be thought of as augmentations but it is unclear if this algorithm will necessarily generalise to those settings because it is unclear if the class of augmentations used here resemble "natural" augmentations in videos.
2. Keeping the hippocampal inspiration aside for a moment, the addition of g and Sim-2 is what distinguishes this algorithm from SimSiam. However, Fig.5. suggests that setting g=I doesn't influence the performance too much - it would be good to know if the continual learning results and other results do require that g is trainable and not I. Similar concern for Sim-2 - what happens when Sim-2 is left out? These ablation analysis are important to understand what makes PhiNet special. Also removing Sim-2 makes the algorithm similar in essence to SimSiam - I'd be curious if this version already outperforms the original SimSiam implementation - this will help us establish a stronger "baseline" for the full PhiNet.

Clarity suggestion:
- In the analytical Section 4, it'd be great if you'd give us an intuition for what phi and gamma in Eq. 1 actually denote - how are they related to g and h, etc. This will truly help tie in the section to the rest of the paper (although it is fine as is too).

**Questions:**

What is the physiological relevance of low weight decay - why is that the important analysis here? (the authors spend a lot of space on it)

---

> ### Author Response · Authors · 2024-11-21
> **Response to Reviewer 1cHn**
>
> We would like to thank the reviewer for the insightful questions and for correctly acknowledging the strengths of our work.
> ## Weaknesses
> > it is unclear if the class of augmentations used here resemble "natural" augmentations in videos
>
> Thank you for your feedback.
> We conducted our experiments under the belief that the data augmentation techniques we applied (such as RandomResizedCrop and RandomHorizontalFlip) closely resemble "natural" augmentations in videos.
> However, we acknowledge that the augmentations may not fully align with natural augmentations. Here, let us clarify the objective of our approach in this work.
> Predictive coding operates under a framework where it (1) processes a single input from a time series and (2) predicts the adjacent input. This approach is expected to offer the benefits of (A) capturing temporal features and (B) stabilizing learning.
> For non-video data, while the framework does not fully satisfy condition (1), it does meet condition (2). Therefore, evaluating predictive coding with existing data augmentations can be reframed as breaking down the framework into components (1) and (2) and initially examining the effectiveness of condition (2).
> The primary focus of our paper is to investigate whether condition (2) implies (B). Testing this framework in more temporally consistent data settings, which would satisfy both conditions (1) and (2), is indeed a necessary step and represents an important direction for future work.
>
> > Fig.5. suggests that setting g=I doesn't influence the performance too much
>
> Thank you for your question.
> As shown in Figure 5, when the batch size is 1024, the accuracy of $g = I$ is lower than that of PhiNet, especially when weight decay is small, resulting in a lower evaluation accuracy (Eval Acc). In addition, we have conducted new experiments on CIFAR-5m and added the results of X-PhiNet with $g = I$ to Table 3.
> From Table 3, it can be observed that even at the optimal weight decay of $2 \times 10^{-5}$, X-PhiNet with $g = I$ achieves an Eval Acc that is 0.6\% lower than the standard X-PhiNet (cos). Moreover, when the weight decay is further reduced to $1 \times 10^{-5}$, X-PhiNet with $g = I$ shows an Eval Acc that is 1.3\% lower than the standard X-PhiNet (cos).
> These results suggest that having $g$ as trainable leads to significantly higher accuracy. We agree that conducting experiments with continual learning is also important, and we plan to include these results in the camera-ready version. Furthermore, the theoretical analysis presented in Section 4 also supports the conclusion that having $g$ as trainable is critical for preventing mode collapse.
>
> > I'd be curious if this version already outperforms the original SimSiam implementation
>
> Thank you for your question.
> If we remove Sim-2 from PhiNet, it becomes equivalent to SimSiam. Therefore, the ablation concerning Sim-2 is effectively comparison between PhiNet and SimSiam. We acknowledge that the difference between PhiNet and SimSiam may not have been clear initially. Based on the advice from Reviewer LRvN, we have improved the presentation by including a figure of SimSiam in Figure 1, making the differences between SimSiam and PhiNet more evident.
>
> > In the analytical Section 4, it'd be great if you'd give us an intuition for what phi and gamma in Eq. 1 actually denote - how are they related to g and h, etc.
>
> Thank you for your valuable feedback.
> In Section4, $\psi$ and $\gamma$ correspond to (one of) eigenvalues of the linear networks h and g, respectively. Intuitively, we can regard $\psi$ and $\gamma$ as “scalarization” of the predictor networks. This formally requires tedious matrix analysis as shown in Appendix. The theoretical analysis is detailed in the Appendix, and we believe that having an intuitive understanding when reading this section would be beneficial. Therefore, we have added the above intuitive explanation to the Appendix.
>
> ## Questions
>
> > What is the physiological relevance of low weight decay - why is that the important analysis here? (the authors spend a lot of space on it)}
>
> Thank you for your question.
> We believe that weight decay is related to synaptic pruning; however, there is still much that remains unknown regarding the strength of pruning in the brain.
> We would like to clarify our initial motivation: it was based on ML research indicating that "non-contrastive learning is sensitive to weight decay and prone to mode collapse." Our primary concern is that this sensitivity and susceptibility to mode collapse would not be ideal even for biological brains as well, since mode collapse would prevent the acquisition of meaningful representations.
> We believe that our proposed PhiNet offers a biologically plausible approach to mitigating mode collapse.
> Connecting this insight with physiological findings would undoubtedly be a fascinating direction for future work.

---

> > ### Comment · Reviewer_1cHn · 2024-11-22
> > **Response to authors**
> >
> > "The primary focus of our paper is to investigate whether condition (2) implies (B)." I am not sure I agree that the substantial transformations you apply here resemble "next-frame" transformations in natural videos. I understand your motivation but I wonder if your approach will actually generalise to the small transformations natural videos bring.
> >
> > "These results suggest that having as trainable leads to significantly higher accuracy." I don't know if 0.6% or 1.3% lower accuracy warrant this conclusion. It helps but it seems we can do very well without it!
> >
> > In general, I think my score is fine - there's something interesting for the field here.

---

### Official Review · Reviewer_LRvN · 2024-11-04

**Soundness:** 3
**Presentation:** 3
**Contribution:** 3
**Rating:** 6
**Confidence:** 4

**Summary:**

The manuscript links the temporal prediction hypothesis observed in the brain's hippocampus-cortex interactions to a novel self-supervised learning structure (PhiNets). PhiNets is largely based on SimSiam but incorporates one more predictor layer and StopGradient module inspired by neuroscience observations. This model demonstrates greater stability and resilience against representational collapse compared to SimSiam and shows superior performance in online and continual learning scenarios. Additionally, the study presents X-PhiNet, an extension incorporating an exponential moving average encoder for improved long-term memory retention in continual learning.

**Strengths:**

- A clever observation between the temporal prediction hypothesis and the StopGradient module, making this architecture more brain-like. However, the definition of temporal steps was poorly articulated and needs more work. (See weakness)
- Rigorously proved the benefits of adding one more predictor layer through the analysis of gradient flow in a linearized case
- Demonstrated improved performance in terms of robustness to weight decay in terms of image classification

**Weaknesses:**

- Presentation of the paper needs more improvement: For example, it would be nice to add the diagram of SimSiam to Figure 1 and this is the main work the current manuscript was based on.
- From the derivation, we know that adding one more layer of signal prediction could prevent from representation collapse. But is this going to slow down the convergence rate? What are the drawbacks of having one more layer of prediction.
- In the performance evaluation plots: Phinet showed sufficient improvement compared to its original version SimSiam but not much improvement compared to other methods such as Barlow twins. And it's unclear why mse loss is better than cos loss in L_{NC}.

**Questions:**

- The weight decay term appeared out of sudden and definite needs more explanation about how it approximates modern optimizers and its relationship with learning rate etc.  because the robustness about decay is where PhiNet showed most improvements in the later numerical experiments.

- The definition of time steps in the network and its link to temporal prediction is not well articulated. I roughy get the general idea how StopGradient could lead to synaptic delays but how are these two linked mathematically is still vague. Does one time step refers to one gradient update? Is gradient update performed in an online manner?

---

> ### Author Response · Authors · 2024-11-21
> **Response to Reviewer LRvN**
>
> We would like to thank the reviewer for the helpful comments and suggestions.
>
> ## Weaknesses
> > it would be nice to add the diagram of SimSiam to Figure 1 and this is the main work the current manuscript was based on.
>
> Thank you for your helpful advice.
> Based on your suggestion, we have added a diagram of SimSiam to Figure 1. We believe this makes the differences between SimSiam and our model more evident. As illustrated in the figure, the architectural novelty of our PhiNet lies in the additional predictor $g$.
>
> > But is this going to slow down the convergence rate? What are the drawbacks of having one more layer of prediction.
>
> Thank you for your question.
> As shown in Figure 6, the use of PhiNet does not result in slower convergence. On the contrary, it often mitigates early-stage instability in learning and allows for more stable progression. A potential drawback of adding one more layer of prediction is the additional memory cost. However, as indicated in Table 17: Comparison of GPU Memory Costs, the memory consumption only increases slightly from 3.26GB to 3.44GB, which we believe is not a significant issue.
>
> > In the performance evaluation plots: Phinet showed sufficient improvement compared to its original version SimSiam but not much improvement compared to other methods such as Barlow twins.
>
> Thank you for your question.
> In the experiments presented in Table 2 on CIFAR-5m and Split C-5m, X-PhiNet achieves an accuracy that is 2\% higher than Barlow Twins, which represents a statistically significant difference. In addition, another advantage of X-PhiNet over Barlow Twins is the reduced computational cost. As shown in Table 18 in the Appendix, when the batch size is as small as 128, Barlow Twins requires nearly three times the training time compared to X-PhiNet.
>
> > And it's unclear why mse loss is better than cos loss in $L_{NC}$.
>
> Thank you for your feedback.
> To clarify, our claim is not that "MSE loss is always superior." In fact, in some experiments using CIFAR-5m, models employing cosine loss can achieve higher performance. Our position is that both models using cosine loss and those using MSE loss are categorized under PhiNet, with the choice of which is better depending on the specific task or setting.
> There remains much to be explored regarding the theoretical analysis of selecting the appropriate loss function for different settings, and it would be a promising future work.
> This is emphasized in the extended limitations section in the appendix.
>
> ## Questions
>
> > definite needs more explanation about how it approximates modern optimizers and its relationship with learning rate etc.
>
> Thank you for your question.
> Weight decay is a standard component used in modern gradient descent and backpropagation methods. In Section 4, weight decay appeared suddenly in the formulation of gradient flow, so we have added a note in Section 3 to clarify that our optimization process includes weight decay.
> Regarding the weight decay introduced in Section 4, we believe that the formulation of gradient flow with weight decay is consistent with discrete gradient descent with weight decay. In the discrete formulation, the weight decay term is indeed proportional to the learning rate; however, when considering the infinitesimal limit of the learning rate in the gradient flow formulation, both sides are divided by the learning rate.
> This point has been further explained at the beginning of Appendix B. Thank you for bringing this to our attention.
>
> >  Does one time step refers to one gradient update? Is gradient update performed in an online manner?
>
> Thank you for your question.
> The "one time step" introduced by the stop-gradient refers to a single gradient update. There are two aspects of time step in our paper:
>
> 1. The time step associated with the next step of the gradient in gradient flow dynamics.
> 2. The time step where the augmented view is considered as the next frame in a time series.
>
> We acknowledge that discussing these two different aspects of time derection may cause confusion. The augmented views are input to PhiNet, creating a temporal lag between inputs, while the temporal lag in the weights arises due to the stop-gradient mechanism.
> Additionally training with CIFAR-5m is conducted in an online learning setting, whereas training with CIFAR10, STL10, and ImageNet follows a mini-batch training.

---

> > ### Comment · Reviewer_LRvN · 2024-11-29
> > **Response to authors**
> >
> > I think the authors have addressed and clarified most of my concerns. Thanks for the efforts. I'll increase my confidence score.

---

### Official Review · Reviewer_pS8F · 2024-11-04

**Soundness:** 2
**Presentation:** 3
**Contribution:** 2
**Rating:** 6
**Confidence:** 3

**Summary:**

In this paper, the authors propose PhiNet, a self-supervised learning (SSL) method partly inspired by the temporal prediction hypothesis and a previous SSL method SimSiam. PhiNet can be seen as an extended version of SimSiam with an additional predictor and a stable encoder which is an exponential moving average of the main encoder. The authors analyzed the learning dynamics of PhiNet in a linear setting and found that the network is less prone to collapse compared to SimSiam. The method is then evaluated using the CIFAR benchmarks and is shown to outperform classical baselines in SSL, especially in online learning and continual learning settings.

**Strengths:**

1. The hypothesized correspondence of the algorithm to the learning and memory mechanisms in the brain is interesting to both ML and neuroscience communities.
2. The authors conducted a detailed analysis of the learning dynamics which nicely explains the advantage over the baseline.
3. There are extensive experiments that clearly illustrate the strengths and weaknesses of the models.
4. The presentation is clear and a lot of details can be found in the appendix.
Overall this is a solid and interesting contribution that bridges predictive coding and SSL algorithms known to the ML community. I thus support acceptance provided that the authors can reasonably address my concerns.

**Weaknesses:**

1. While this is understandable for a more brain-inspired algorithm, there is little performance gain on classical SSL benchmarks.
2. The link from the model and task setting to the temporal predictive coding idea is not that strong, see below.

**Questions:**

1. One major concern is that the proposed method is not really doing "temporal predictive coding", instead, the predictor is predicting the embedding of an augmented image. The way to introduce the model is thus somewhat misleading. It also seems more natural to use some video datasets if the goal is to build a model for temporal predictive coding. I wonder if the authors can explain more about the motivations here.
2. When connecting X-PhiNet to the brain as in Fig 1(c), we are basically assuming that layers II and III in EC share the same encoder and the NC encoder is EMA of the one in EC. I wonder if the authors can provide more rationale for these assumptions.
3. While there are some qualitative comparisons of the learning dynamics between PhiNet and SimSiam, I think the result would be stronger if there were also quantitative comparisons, e.g., something showing that more random seeds would result in collapse for SimSiam than for PhiNet.
4. For empirical performance, the errorbar is missing from Fig 5 and Fig 7 so it's a bit hard to measure the margin of error.
5. When comparing models on online learning and continual learning benchmarks, it seems that the EMA encoder is critical, but at least for continual learning, this is not very surprising since EMA essentially prevents the representations from drifting away too much. As baselines for comparison here (POYO, SimSiam, etc) are not designed for continual learning, it might make sense to include more baselines from the field of continual learning, otherwise, the claim that X-PhiNet particularly excels at online and continual learning seems a bit unfair.

---

> ### Author Response · Authors · 2024-11-22
> **Response to Reviewer pS8F**
>
> We thank the reviewer for your positive evaluation and constructive feedback on our paper.
>
> ## Weaknesses
>
> > While this is understandable for a more brain-inspired algorithm, there is little performance gain on classical SSL benchmarks.
>
> Thank you for your feedback.
>
> It is true that the performance gain observed with CIFAR10 appears limited. However, a more pronounced performance difference emerges in CIFAR-5m and continual learning scenarios. We believe that CIFAR-5m and continual learning, rather than CIFAR10, represent settings that are more natural both biologically and from an engineering perspective.
> Traditionally, self-supervised learning experiments have focused on datasets such as CIFAR10 and ImageNet, often utilizing mini-batch training and requiring a substantial number of epochs—800 epochs for CIFAR10, for example—far exceeding those used in supervised learning. This extensive training reduces biological plausibility, as animals inherently learn through online processes.
> In addition, recent advancements in language models demonstrate the effectiveness of training on large-scale web data with a reduced number of epochs, sometimes as few as one. This progress underscores the importance of developing similar single-epoch training strategies for image-based self-supervised learning. Our experiments with CIFAR-5m revealed substantial differences between methods in online learning settings, distinctions that are not evident with CIFAR10.
>
> ## Questions
>
> > One major concern is that the proposed method is not really doing "temporal predictive coding", instead, the predictor is predicting the embedding of an augmented image...I wonder if the authors can explain more about the motivations here.
>
> Thank you for your question.
>
> Predictive coding operates within a framework where (1) it processes a single input from a time series and (2) predicts the next input signal. The advantages of this approach are expected to include (A) capturing temporal features and (B) stabilizing learning. For non-video data, while the framework does not fully satisfy condition (1), it does meet condition (2). Therefore, testing predictive coding with non-video data can be reframed as breaking down the framework into components (1) and (2), focusing initially on evaluating the effectiveness of condition (2).
> The scope of our current paper can thus be described as examining whether condition (2) implies (B). Validation of predictive coding on video data, incorporating both conditions (1) and (2), is indeed an essential direction for future work, which we intend to pursue.
>
> > When connecting X-PhiNet to the brain as in Fig 1(c), we are basically assuming that layers II and III in EC share the same encoder and the NC encoder is EMA of the one in EC. I wonder if the authors can provide more rationale for these assumptions.
>
> Thank you for your question.
>
> SimSiam and the temporal predictive hypothesis differ primarily in whether they incorporate an additional predictor $g$, assuming we accept that the EC layers are modeled by the same encoder. Our research focuses on evaluating whether a biologically plausible recurrent structure involving $g$ has a mathematically significant effect.
> Given that Layers II and III receive inputs from two images at different time steps, assuming that these layers share the same encoder is a natural assumption.
> Furthermore, using EMA to model NC layer is a common approach in the context of CLS theory[McClelland 1995, Pham 2021]. However, it should be noted that this is merely one of the simpler methods for modeling slow and fast learning systems.
> Since many aspects of the biological brain remain unclear, further investigation is necessary to validate these assumptions.
>
> [McClelland 1995] J. L. McClelland, B. L. McNaughton, and R. C. O’Reilly. Why there are complementary learning
> systems in the hippocampus and neocortex: insights from the successes and failures of connectionist
> models of learning and memory. Psychological Review, 1995.
> [Pham 2021] Q. Pham, C. Liu, and S. Hoi. DualNet: Continual learning, fast and slow. NeurIPS, 2021.

---

> > ### Author Response · Authors · 2024-11-22
> >
> > > I think the result would be stronger if there were also quantitative comparisons, e.g., something showing that more random seeds would result in collapse for SimSiam than for PhiNet.
> >
> > Thank you for your helpful suggestions.
> > The results in Figure 5 (now Figure 6) have been updated to include error bars based on multiple seeds. From these results, it can be observed that SimSiam exhibits unstable training when the batch size is 1024 and the weight decay is small, resulting in large variability in accuracy across different seeds.
> > While it is challenging to definitively distinguish whether mode collapse occurs during training, we believe that the large variability in the accuracy of SimSiam across seeds and its overall lower accuracy suggest that "more random seeds would result in collapse".
> >
> > > For empirical performance, the errorbar is missing from Fig 5 and Fig 7 so it's a bit hard to measure the margin of error.
> >
> > Thank you for your helpful advice.
> > Following your suggestion, we have added error bars to Figure 5 (now Figure 6). However, due to time constraints, we have not yet been able to include error bars for the continual learning results in Figure 7 (now Figure 8).
> > For the error bars of the continual learning results, please refer to Tables 5, 6, and 7 in the Appendix, where error bars are provided. While these tables do not include a sweep over weight decay, they demonstrate the statistically significant differences achieved by X-PhiNet.
> >
> > > As baselines for comparison here (POYO, SimSiam, etc) are not designed for continual learning, it might make sense to include more baselines from the field of continual learning
> >
> > Thank you for your feedback.
> > As you remark, SimSiam and BYOL are not designed for continual learning. Although BYOL also employs EMA, its performance in online and continual learning settings is not as strong as X-PhiNet, indicating that EMA does not guarantee high performance. Furthermore, RM-SimSiam, which we employ as a baseline in our comparisons, is designed for continual learning and also utilizes EMA. Despite this, X-PhiNet outperforms RM-SimSiam.
> > In addition to RM-SimSiam, another self-supervised method for continual learning is CasSSLe. While it would indeed be valuable to include CasSSLe[Fini2022] as a baseline for comparison, the limited rebuttal period make it challenging to conduct additional experiments at this time. A more detailed analysis in the context of continual learning would be a future work.
> > While you may feel that there is a lack of strong baselines for continual learning, to the best of our knowledge, within the context of non-contrastive learning, most research has focused on data-structure-based approaches such as replay strategies [Madaan 2022].
> >
> > [Fini2022] Fini, E., da Costa, V. G. T., Alameda-Pineda, X., Ricci,
> > E., Alahari, K., and Mairal, J. Self-supervised models are continual learners. CVPR, 2022.
> > [Madaan 2022] D. Madaan, J. Yoon, Y. Li, Y. Liu, and S. J. Hwang. Representational continuity for unsupervised continual learning. ICLR, 2022.
> >
> > If our response satisfies the reviewer, we appreciate it if you would consider adjusting their score accordingly.

---

> ### Comment · Reviewer_pS8F · 2024-11-26
>
> Thanks for the thoughtful response.
> > The scope of our current paper can thus be described as examining whether condition (2) implies (B).
>
> I see the point here, and I hope the manuscript clarifies this. But without the temporal component, I think there is still a slight mismatch between the motivation and the actual method.
> > While it is challenging to definitively distinguish whether mode collapse occurs during training, we believe that the large variability in the accuracy of SimSiam across seeds and its overall lower accuracy suggest that "more random seeds would result in collapse"
>
> I think without a more detailed analysis of the empirically learned representations this argument is a bit handwavy, but I understand that this might be hard to do in the short rebuttal period.
> I want to thank the authors for addressing most of my concerns. I will keep the score since the original score already assumes these concerns are solved.

---

### Meta-Review · Area_Chair_AQVo · 2024-12-16

**Metareview:**

This paper presents a novel self-supervised learning technique, loosely inspired by work on the hippocampus in neuroscience, called PhiNet. Similar to SimSiam, PhiNet take an input, generates two augmentations. But, unlike PhiNet, it makes a prediction with both augmentations (one of them via two steps of processing), and then compares these to an encoding of the original input. This is supposed to be analagous to the CA1-CA3-EC circuit in the medial temporal lobes.

The authors claim that PhiNet is able to learn useful representations more stably than SimSiam. They also claim that it can adapt more quickly in online learning scenarios. Finally, they claim to have a version of the model using an exponential moving average that is even better on continual learning tasks.

The strengths of the paper are that the idea and its link to neuroscience are interesting and the evaluations are robust and convincing. The weaknesses are that the advance over existing SSL methods is not that great, and some aspects of clarity in the paper were lacking.

Given these considerations, and the final scores (6,6,8,8) a decision of accept (poster) was reached.

**Additional Comments On Reviewer Discussion:**

The discussion was constructive, and the authors assuaged the reviwer concerns enough to warrant acceptance.

---

### Decision · Program_Chairs · 2025-01-22

Accept (Poster)